


# Remote sensing of exceptional winter aerosol pollution events and representativeness of the surface – column relationship over Paris metropolitan area

Alexandre Baron, Patrick Chazette, Julien Totems

Laboratoire des Sciences du Climat et de l'Environnement, LSCE-IPSL, CEA-CNRS-UVSQ, UMR 8212, Gif-sur-Yvette, France

*Correspondence to*: Alexandre Baron (alexandre.baron@lsce.ipsl.fr)

**Abstract.** In this study an optical parameter derived from lidar measurements is found to be relevant to monitor the evolution of near-surface particulate concentrations. This highlights the opportunities offered by future spaceborne lidar missions in air

quality assessment on a global scale. This work is carried out following a dedicated field campaign in the Paris area (France) during winter 2016-2017, from 1st November to 31st January. Two of the most intense winter aerosol pollution events occurring over the last decade were sampled using a ground-based $N_2$-Raman. The lidar operated continuously at the wavelength of 355 nm, favourable to the measurement of submicron aerosols mainly linked to traffic emissions. The data analysis uses the synergy between ground-based and spaceborne lidar observations, and data from the air quality monitoring network Airparif. The first

severe aerosol pollution event occurred on 1st December 2016; it concerned a circular area of 250 km in diameter around Paris with maximum $PM_{10}$ ($PM_x$ is the mass concentration of particles with an aerodynamic diameter smaller than x μm) values of $121\pm63\,\mu g\,m^{-3}$. The second event took place from 21st to 22nd January which covered all of Western Europe, with maxima of $PM_{10}$ ($156\pm33$ μg $m^{-3}$) and aerosol extinction coefficient (AEC) between 0.6 and 1 $km^{-1}$, within the winter atmospheric boundary layer. These two major aerosol pollution events share very low boundary layer height, down to 300 m above ground

level. However, they did not take place in the same weather condition; moreover, they are associated with significantly different lidar ratios: $72\pm15$ sr and $56\pm15$ sr, respectively in December and January. Such results are consistent with available spaceborne lidar data ($70\pm25$ sr) and values found in the literature. During these two events, the continuous temporal evolution of the aerosol extinction coefficient allows us to investigate the representativeness of optical parameters found in the planetary boundary layer to assess surface aerosol concentration. No one-to-one relationship between the aerosol optical thickness (AOT)

and $PM_{2.5}$ values stands out within our study. In contrast, the maximum lidar-derived aerosol extinction coefficient found within the planetary boundary layer is identified as a consistent variable to assess the evolution of ground aerosol concentration.

*Keywords: lidar, aerosols, spaceborne, AOT, $PM_{2.5}$, winter, urban, pollution.*



## 1 Introduction

According to the report of the Organization for Economic Co-operation and Development (OECD), particulate matter (PM) is one of the main anthropic contributors affecting human health and agriculture (OECD, 2016). Indirectly, this pollution induces substantial economic tolls: the same report claims that global healthcare costs related to air pollution rose to USD 21 billion
in 2015 and projections reach up to USD 176 billion by 2060. Furthermore, aerosols are responsible for a significant decrease in life expectancy in large urban and industrial areas (IIASA, 2000). Whereas they represent a small portion of the cumulative exposure of an urban dweller, severe aerosol pollution events are known for their important short-term impact on human health and especially excess mortality in at-risk populations (Hogg and Van Eeden, 2009). Once advected in the atmosphere, their effects on health and climate (IPCC, 2013) grow from regional to global scale.

Places where inhabitants are the most concerned and vulnerable to particulate pollution are megacities (Molina and Molina, 2004). Thus, several studies were performed in such conurbations in order to investigate the aerosol impacts on air quality and climate, such as in Mexico in 2016 with the Megacity Initiative: Local and Global Research Observations (MILAGRO, Molina et al., 2010) or in California during California Nexus (CalNex, e.g. Hersey et al., 2013). International field experiments were also carried out in Europe as well as in the United Kingdom, with the M25 (ring-road around London) experiment (McMeeking
et al., 2012), or in southern France with the Expérience sur Site pour Contraindre les Modèles de Pollution atmosphérique et de Transport d'Emissions (ESCOMPTE, Cros et al., 2004). Nevertheless, air quality remains a great challenge to resolve in the future as the European Commission estimates that 90% of EU citizens are regularly exposed to air pollutant levels above the World Health Organization (WHO) guidelines (European Commission, 2015).

Being one of the densest urban areas in Europe, with more than 18% of the French population concentrated in 2.2% of its
metropolitan territory (Pereira et al., 2013; Ile de France Prefecture, 2017), the Paris megacity (Paris city limits extended to all its suburbs in the Ile-de-France region) is often impacted by air pollution issues. Several field campaigns have been conducted in the Paris Area: the Etude et Simulation de la QUalité de l'air en Ile-de-France project (ESQUIF, Vautard et al., 2003; Chazette et al., 2005; Hodzic et al., 2006), LIdar pour la Surveillance de L'AIR (LISAIR, Raut and Chazette, 2007) and The Megacities: Emissions, urban, regional and Global Atmospheric POLlution and climate effects, and Integrated tools for
assessment and mitigation project (MEGAPOLI, e.g. Von Der Weiden-Reinmüller et al., 2014; Freney et al., 2014). These campaigns allowed to assess the environmental impacts of the Paris megacity (Skyllakou et al., 2014) mainly during summer time and improved the predictability (Beekmann, 2003; Tombette et al., 2009; Royer et al., 2011b) and the source apportionment of aerosol pollution events (Sciare et al., 2010; Crippa et al., 2013a; Bressi et al., 2013, 2014; Pikridas et al., 2015). So far, existing studies of winter aerosol pollution events (APEs) in Paris were mainly derived from modelling
(Bessagnet et al., 2005; Crippa et al., 2013b; Beekmann et al., 2015). They demonstrate that in weather-blocking conditions and with a cold surface, the PBL top height, and therefore the dilution capability along the altitude, are low, resulting in forecast uncertainties (e.g. Steeneveld, 2011).





The use of *in situ* sounding, or for more vertical and temporal resolutions, lidar remote sensing, can solve this problem. Establishing a relationship between particle concentration at ground level and optical observations within the PBL may enhance the predictability of air pollution peaks and improve the assessment of air quality on a continental scale (Wang et al., 2013). Many authors seek to derive such a relationship between $PM_{2.5}$ and optical observations, mainly performed from

satellites, as shown in the recent review study of Chu et al. (2016).

Hence, the main purpose of this paper is to investigate the link between ground-based aerosol measurements and particles trapped within the winter planetary boundary layer (PBL) as observed by a ground-based dinitrogen Raman lidar ($N_2$-lidar). This study is based on a specific field campaign performed during the most severe winter aerosol pollution events (APEs) above the Paris area. The need for such a study to enhance the accuracy of $PM_{2.5}$ assessment using surface aerosol extinction

is clearly identified in the conclusion of Toth et al. (2014), opening the way for more accurate assessments of air quality on a global scale with the upcoming new generation of spaceborne lidar, such as those carried onboard ADM-Aeolus (Flamant et al., 2008) and further the European spaceborne mission EarthCARE (Illingworth et al., 2015).

To draw a relationship between surface concentration of aerosol and their optical properties within the PBL, the following approach is used in this paper: i) lidar measurements are inverted in order to retrieved aerosol optical properties, whose the

lidar ratio (LR) and the aerosol extinction coefficient (AEC), ii) the linear particle depolarisation ratio (PDR) is then assessed and used with the LR to identify the aerosol typing, iii) the meteorological situation is checked to ensure an unique origin of pollution aerosol over the Paris area, iv) spaceborne observations are used when available to corroborate the meteorological analysis and give a regional view of pollution plume and, v) the links between the aerosol optical properties within the PBL and the ground-based particulate matter is studied.

Section 2 presents the instruments and datasets involved in the study. Passive and active remote sensing measurements are mainly used, completed by meteorological model outputs and the dataset of the Airparif air quality measuring network. We highlight the most polluted winter days of the past decade in Section 3 from archived ground-based $PM_{10}$ measurements and analyse the associated weather situations. In Section 4 we analyse in depth the measurements performed during winter 2016/2017, which included those of the $N_2$-Raman ground-based lidar. This period was associated with two exceptional APEs

whose characteristics are described in Section 5: we take advantage of the entire lidar dataset of these APEs. It covers a wide range of aerosol load, from pollution free days to severely polluted days. These data are analysed in terms of correlation between ground level and aerosol optical properties within the PBL using Airparif particle matter (PM) measurements and lidar vertical profiles, respectively. Finally, Section 6 concludes this paper by summarizing the main results.

## 2 Methodology and tools

This section presents the instruments and tools used during and following the field campaign held from November 2016 to the end of January 2017.



## 2.1 Ground-based lidar measurements and analysis

The compact 355 nm Lidar for Automatic Atmospheric Surveys Using Raman Scattering (LAASURS) was deployed close to the centre of Paris on the rooftop of the Paris Sorbonne University Jussieu Campus (48° 50′ 50" N, 2° 21′ 20" E), ~100 m above mean sea level (AMSL). The LAASURS has been successfully involved in former field campaigns as in Chazette and
Totems (2017), Dieudonné et al. (2015, 2017), and it is extensively described in Royer et al. (2011a).

### 2.1.1 Technical characteristics

The LAASURS has been developed at LSCE for ground-based or airborne aerosol remote sensing in the field. The reception is composed of three channels, two using elastic scattering filtered at 354.7±0.1 nm and separated into parallel and perpendicular polarizations of light with respect to the laser emission, as well as a third channel using the inelastic nitrogen
vibrational Raman scattering induced by the laser and filtered at 387.6±0.1 nm (Chazette et al., 2016). The three channels provide measurements from their short overlap distance of ~150 m up to 20 km. The overlap factor is calculated during night time using a horizontal line of sigh as described in Chazette et al., (2007) assuming a homogeneous layer of aerosols from the emission to 1.5 km horizontal distance. In the lowermost 150 m, the uncertainties induced by the overlap factor do not permit to assess aerosol optical properties with a sufficient level of confidence. The emission is provided by an Ultra® Nd:YAG laser
manufactured by Quantel delivering 6-7 ns pulses of 30 mJ at a 20 Hz frequency. The initial vertical and temporal resolutions of the lidar are respectively 0.75 m and 50 s (1000 laser shots averaged). More characteristics are given in Table 1 and the casing enclosing the lidar is presented in Figure 1.

### 2.1.2 Inversion of lidar profiles

To obtain a sufficient signal to noise ratio (SNR) from the $N_2$-Raman channel during daytime, the vertical resolution is
downgraded to 15 m and lidar profiles are averaged over 10 minutes. Two methods can be used to retrieve the aerosol optical parameters: i) A synergy between the elastic channels and a sun-photometer as in Chazette (2003), Pahlow et al.(2006), Raut and Chazette(2007) and Cuesta et al.(2008); or, ii) the use of a $N_2$-Raman channel as described in Russo et al. (2006), Ansmann et al. (2008) or Royer et al. (2011a). In this work, the presence of the $N_2$-Raman channel of the LAASURS makes it preferable to use the second approach detailed by Royer et al. (2011a) and Chazette et al. (2017). Readers can refer to these articles. In
the following inversion process, the extrapolation of AOT measurements from the $N_2$-Raman wavelength to the elastic wavelength assumes a constant Ångström exponent for the particles in the atmospheric column. The Ångström exponent is derived from the AErosol RObotic NETwork database (AERONET, http://aeronet.gsfc.nasa.gov/) for the Paris site. This assumption is consistent as all aerosols are concentrated in the shallow PBL with low horizontal advection of aerosol plumes. During daytime, the lidar-derived AOT is checked against the one measured by the sun-photometer of the AERONET Paris
site at concomitant times. The AOTs, combined with the elastic channel, lead to the retrieval of the aerosol extinction and backscatter coefficient profiles (AEC and ABC, respectively), and to their ratio, also called LR. The LR may vary in the





atmospheric column since different types of aerosols can be present. However, in the particular case of this work, with most of the particles trapped close to ground level in a winter PBL, the assumption of an equivalent LR for the entire column is justified. When the aerosol load is sufficient (ABC 5% above the molecular backscatter coefficient), the linear particulate depolarization ratio (PDR) is computed as in Chazette et al. (2012).

5  ### 2.1.3 Uncertainties

The different sources of uncertainties are discussed in Royer et al. (2011a) where the authors used a Monte Carlo method applied to the direct-inverse model of the lidar to obtain its error budget. These uncertainties are strongly dependent on the SNR. The SNR encountered in these measurements (signal originating from the lower troposphere: altitude < 2-3 km AMSL) remains greater than 10. In these conditions, the relative uncertainty on the $N_2$-Raman-derived AOT is less than 2%. The

10  relative uncertainties on the LR and AEC vary from 7% and 3%, respectively, for AOT greater than 0.5. They increase to 23% and 13%, respectively, for AOT ~ 0.1 (see Table 2 in Royer et al. (2011a)). We can consider a standard deviation of 10 sr on the LR for AOT ≳ 0.2. Relative uncertainties on the PDR retrieval are discussed in Dieudonné et al. (2015; 2017) and are of the same magnitude as those associated to the LR. For small PDR values (< 5%), the absolute error is between 1 and 2 %.

**Table 1: LAASURS characteristics**

| | |
|---|---|
| **Emission wavelength** | 354,7 nm |
| **Laser energy** | 30 mJ |
| **Pulse duration** | 6-7 ns |
| **Shooting frequency** | 20 Hz |
| **Emission lens diameter** | Ø 50 mm |
| **Reception lens diameter** | Ø 150 mm |
| **Field of view** | 2 x 0,67 mrad |
| **Complete overlap distance** | 150m to 200 m |
| **Elastic channels wavelength** | 354.7±0.1 nm |
| **Raman $N_2$ channel wavelength** | 387.6±0.1 nm |
| **Detector** | Photomultiplier |
| **Acquisition mode** | Analog and photon count |
| **Acquisition frequency** | 200 MHz |
| **Spatial resolution** | 0.75 m to 15 m |



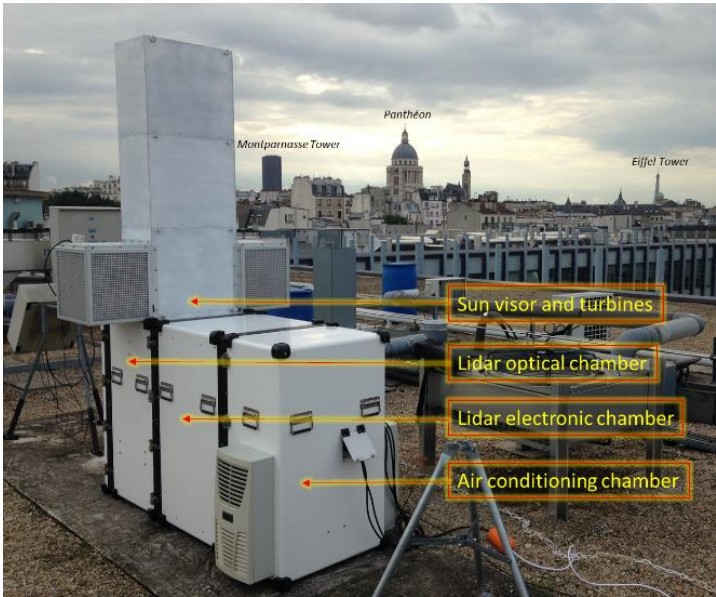

**Figure 1: LAASURS system on the Paris Sorbonne University rooftop**

## 2.2 Spaceborne instruments

### 2.2.1 MODIS

5   On-board both Terra and Aqua satellites, the Moderate Resolution Imaging Spectroradiometer (MODIS) (Salomonson et al., 1989; King et al., 1992; Remer et al., 2005) is composed of 36 spectral bands ranging from 400 nm to 1440 nm. Its swath is 110° (2330 km) and the resolution at ground level varies from 250 m to 1000 m, depending on the band used. Here we use the AOT at 550 nm included in the collection 6 (C6) deep blue aerosol products MOD04_L2 and MYD04_L2 (Levy et al., 2013). The predicted uncertainty over land on the AOT at 550 nm remains as in collection 5 (C5): ±0.05+0.15 AOT (Levy et al.,

10  2010).

### 2.2.2 CALIOP

Launched in April 2006 to be part of the A-Train (Stephens et al., 2002), the Cloud Aerosol Lidar and Infrared Pathfinder Satellite Observations (CALIPSO) is a satellite carrying a backscatter lidar for atmospheric observations purposes. The spaceborne Cloud-Aerosol LIdar with Orthogonal Polarization (CALIOP) is composed of a diode-pumped Nd:YAG emitting

15  110 mJ linearly polarized pulses at a repetition rate of 20.25 Hz at both 1064 nm and 532 nm wavelengths (Winker et al., 2003). Horizontal and vertical resolutions are respectively 333 m and 30 to 60 m. Here we take advantage of its level 2 V4.20 operational products (Mamouri et al., 2009) when the CALIPSO track passes above the Paris Area (within 200 km).




### 2.2.3 CATS

The spaceborne lidar Cloud-Aerosol Transport System (CATS, https://cats.gsfc.nasa.gov/) was operating on-board the International Space Station (ISS) from January 2015 to October 2017. The data used in this work come from the mode 7.2 HSRL Demo of the CATS mission. Namely, this acquisition mode uses the backscattered light emitted at 532 and 1064 nm and the depolarization of the 1064 nm channel. In this study, we use the operational product of version L2O_V3-00 (NASA, 2017) with aerosol typing based on lidar ratio considerations. It comes as a complement to CALIOP data, strengthening the credibility of their concomitant results.

Note that CALIOP and CATS have a different typology for aerosol subtyping and associated lidar ratio. Readers can find in (Kim et al., 2018) the selection algorithm used for CALIOP data version 4 and in (Yorks et al., 2015) the theoretical basis of CATS algorithm. For the *polluted continental/ smoke* aerosol subtype, CALIOP and CATS give a LR of 70±25 sr and 65 sr, respectively.

### 2.3 Ground-based networks and model outputs

### 2.3.1 AERONET

The AErosol RObotic NETwork (AERONET, https://aeronet.gsfc.nasa.gov/) is a global network of automatic sun-photometers (Holben et al., 1998). The sun- and sky-scanning provide long-term and continuous monitoring of aerosol optical, microphysical, and radiative properties. The current processing algorithms are in their third version and composed of three quality levels: 1.0 (unscreened), 1.5 (cloud-screened and quality controlled) and 2.0 (quality-assured). The uncertainty on AOT is 0.01 for wavelength $\lambda > 440$ nm (Holben et al., 1998) and up to 0.02 for other wavelengths (Dubovik et al., 2000), but additional bias may appear in presence of thin unscreened cirrus (Chew et al., 2011). To prevent this, we use level 2.0 products and we highlight the presence of clouds in lidar vertical profiles.

### 2.3.2 Airparif

Airparif's mission is to monitor the air quality in the region of Paris, and to inform citizens and authorities if regulatory thresholds on several gaseous or PM pollution are exceeded. These thresholds are taken from two EU directives (n° 2008/50/CE and 2004/107/CE) transposed into French law. We consider the annual average limits for $PM_{10}$ and $PM_{2.5}$, (40 μg m$^{-3}$ and 25 μg m$^{-3}$, respectively) as well as the information and the alert thresholds for $PM_{10}$ (50 μg m$^{-3}$ and 80 μg m$^{-3}$, respectively).

The ground-based stations included in the Airparif network are divided into two main categories: the traffic and the background stations. Then, the background stations are split in three sub-types: urban, sub-urban and rural stations according to their geographical location in the Paris region. Here, only these background stations are considered for the winter months of December, January and February. Fourteen stations measure $PM_{10}$ whereas only nine measure $PM_{2.5}$ in 2017; these numbers have varied over the past years and are taken into account when calculating uncertainties on the spatial average. Only dry PMs





are measured using a Tapered Element Oscillating Microbalance (TEOM), as the sampling is performed through a warmed inlet. On a daily average, the uncertainty associated with this measure is within 9% - 16% for $PM_{2.5}$ and 9% - 21.6% for $PM_{10}$ over the studied period (http://www.airparif.asso.fr/telechargement/telechargement-statistique).

### 2.3.3 ECMWF ERA5 reanalysis

Used for a better understanding of the weather situation from a synoptic point of view, the meteorological data in this paper comes from the European Centre for Medium-Range Weather Forecasts (ECMWF) and more precisely their fifth generation of atmospheric reanalyses of the global climate: ERA5 (ECMWF, 2017). We use reanalyses with a spatial resolution of 0.25° latitude x 0.25° longitude on 37 pressure levels, which are produced every hour.

## 3 Major winter pollution events of the past decade (2007-2017)

### 3.1 Identification from ground-level in situ sampling

The time series of $PM_{10}$ between December 2007 and February 2018 (www.airparif.asso.fr/en) over the Ile-de-France region are investigated. The identification of the main APEs is performed in three steps: i) as official thresholds are defined daily, a daily average is computed for each background station; ii) we select the days during which at least one background station exceeds the $PM_{10}$ information threshold of 50 µg m$^{-3}$ or the alert threshold of 80 µg m$^{-3}$. iii) To single out the most polluted,

regional-scale events of the past decade, we perform a spatial average on all the background stations and select the days with a mean $PM_{10}$ above 80 µg m$^{-3}$.

Figure 2 gives the histogram for APEs exceeding the information threshold. The light (dark) colours represent the occurrence of days with at least one station exceeding 50 µg m$^{-3}$ (80 µg m$^{-3}$). Such a selection yields an overview of polluted days in winter during the last decade. We count 136 (27) days with at least one station exceeding the information (alert) threshold. Figure 2

also shows a slight improvement of the air quality in the Paris metropolitan area in winter over the last decade. The frequency of pollution threshold overruns tends to decrease over the years. Yet, winter 2016/2017 stands out as one of the most polluted of the past decade. In opposition, the previous and following winters, i.e. 2015/2016 and 2017/2018, have very low pollution levels. Despite the increasingly coercing political measures to improve air quality, this multi-annual variability is a result of the deep sensitivity of air-quality to the occurrence of a strong and durable anticyclonic situation.

When considering the spatial averaging, 8 days of aerosol pollution are highlighted, split into the four different episodes presented in Table 2. The first two episodes (during winter 2007/2008 and winter 2008/2009) seem to have been the most severe: each time, the daily spatial average surpassed 110 µg m$^{-3}$ in Metropolitan Paris during two consecutive days. During the winter of 2016/2017, we count two extremely polluted days belonging to two distinct APEs. Both are sampled by the ground-based lidar. As will be presented in Section 4, they share significant AEC levels and shallow PBLs. The first event

begins on 30$^{th}$ November 2016 and ends on 2$^{nd}$ December late in the evening. The second event takes place in January 2017: it begins on 20$^{th}$ January and ends during the night of 23$^{rd}$-24$^{th}$ January, with a peak on 22$^{nd}$ January. According to ERA5




reanalyses, the meteorological patterns are similar over the 8 days: surface pressure above 1015 hPa, temperature close to 0°C, relative humidity around 80% and very low wind speed ($< 3$ m s$^{-1}$) within the PBL (Table 2).

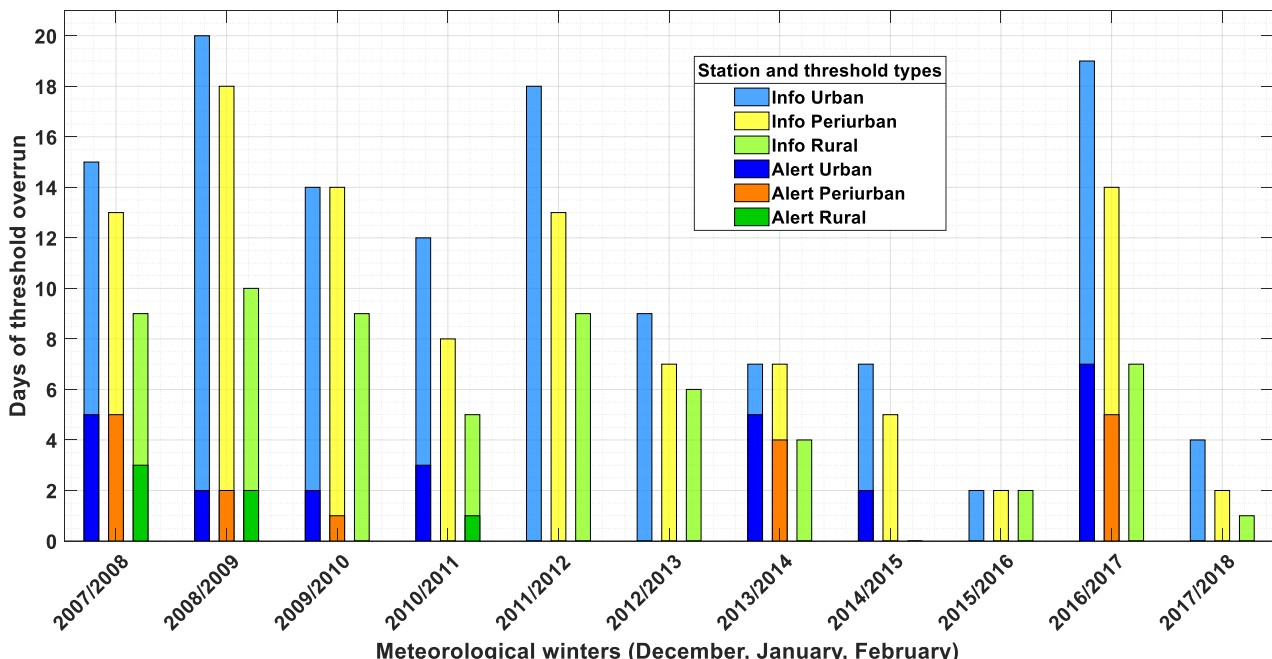

**Figure 2: Diagram representing the number of days comprising at least one station where a threshold is exceeded (light colours for information and dark colours for alert) for each winter of the past decade. The stations taken into account are only background ones. The station typing (urban [light / deep blue], suburban [yellow / orange] and rural [light / deep green]) is conserved to appreciate the spatial extent of a typical polluted day.**

**Table 2: The 8 most severely polluted days of the past decade. For each day we give both PM$_{2.5}$ and PM$_{10}$ measured at ground level (Airparif network) in the format Max/Mean/Min where: Max and Min are the maximum and minimum value measured at a given background station during the day and Mean is the daily average over all background stations. Meteorological parameters (ECMWF ERA5) at ground-level are also given: pressure (P), temperature (T), relative humidity (RH), wind speed (WS) and wind direction (WD) with the standard deviation associated to the daily average.**

| Event # (year) | 1 (2007) | | | | 2 (2009) | | 3 (2016) | 4 (2017) |
|---|---|---|---|---|---|---|---|---|
| Date | 21$^{st}$ Dec | 22$^{nd}$ Dec | 23$^{rd}$ Dec | 24$^{th}$ Dec | 10$^{th}$ Jan | 11$^{th}$ Jan | 1$^{st}$ Dec | 22$^{nd}$ Jan |
| PM$_{10}$ (µg m$^{-3}$) | 151/82/35 | 204/88/26 | 215/131/86 | 182/121/58 | 223/111/60 | 238/123/68 | 241/97/24 | 171/82/39 |
| PM$_{2.5}$ (µg m$^{-3}$) | - | - | - | - | 176/100/54 | 208/120/55 | 159/64/19 | 156/64/32 |
| P (hPa) | 1016±1 | 1015±1 | 1018±1 | 1017±1 | 1017±1 | 1019±1 | 1019±2 | 1015±1 |
| T (°C) | -1.7±3.2 | -0.8±3.5 | 0.1±3.2 | -0.7±3.5 | -6.9±3.5 | -3.8±4.8 | -0.7±3.5 | -3.2±3.4 |
| RH (%) | 83±9 | 81±9 | 85±9 | 87±9 | 86±8 | 82±12 | 84±11 | 83±9 |
| WS (m s$^{-1}$) | 2.3±0.2 | 2.1±0.4 | 1.3±0.8 | 2.2±0.6 | 1.4±0.8 | 2.8±0.5 | 1.6±0.8 | 1.8±0.4 |
| WD (°) | 126±10 | 155±22 | 217±97 | 180±12 | 140±92 | 180±8 | 274±27 | 99±15 |





## 3.2 Favourable weather conditions

Over the Paris area, APEs occur when weather conditions favour northeast advection in the lower and middle troposphere towards the Ile-de-France region (e.g.Chazette and Royer, 2017), or in presence of a weather-blocking situation (e.g. Menut et al., 1999; Bessagnet et al., 2005; Petit et al., 2017), particularly in winter.

### 3.2.1 Synoptic situation

The analysis of large-scale meteorological patterns before and during each of the four afore mentioned events helps to understand how a severe APE can settle in. The common denominator is the perturbation of the usual oceanic wind regime by a high-pressure system. APEs of 2007, 2009 and 2017 share a similar establishment process: a high-pressure system descending from high latitudes blows air from Eastern Europe and settles above Central Europe during a few days. While this meteorological episode lasts, the Azores anticyclone combines with the existing high in central Europe to form a vast high-pressure system with very little wind, as shown in Figure 3a. A low coming from the South for 2009 and 2017, or the West for 2007, finally weakens the high and ends meteorological conditions favourable to an APE.

Figure 3b shows the weather situation that occurred in early December 2016 at the 975-hPa level (within the PBL). It slightly differs from the other three major APEs in terms of location and orientation of its high-pressure system. Centred between Ireland and England, this transient but strong anticyclone blocks air masses of the Paris Area and more broadly Northern France and Southern England, thus nullifying winds. During the night from 2nd to 3rd December, the deep low off the cost of Portugal weakens the high, making it go back to high latitudes. Located at the edge of the remaining high, the Paris Area is seeing the return of winds and a dilution of its air pollution after 3rd December 2016.






**Figure 3: Weather situation on 22$^{nd}$ January 2017 at 1200 UTC (a) and on 1$^{st}$ December 2016 at 1200 UTC (b). The geopotential altitude at a 975-hPa level are also given in white lines. The wind velocity and direction are given by black arrows. Maximum wind speed is 27.8 m s$^{-1}$ (31.6 m s$^{-1}$) at 57.75°N 28.25°W (62.25°N 39.5°W) on the top (bottom) map. Minimum wind speed is 0.02 m s$^{-1}$ (0.03 m s$^{-1}$) at 50.5°N 6.5°E (46.25°N 7.75°E) on the top (bottom) map.**

### 3.2.2 Local winds

At ground level, ERA5 reanalyses from ECMWF show how wind patterns behave in Paris during an APE. We consider the grid point of 0.25° x 0.25° which includes central Paris. Figure 4 displays the hourly wind speed of the four days comprising the APE of December 2016. We note that two days before the event (28$^{th}$ and 29$^{th}$ November), winds are above 4 m s$^{-1}$ and coming from the northeast (Figure 4), whereas for the two polluted days on 30$^{th}$ November and 1$^{st}$ December, winds remain below 3 m s$^{-1}$ and have no privileged direction.

Likewise, for the three cases of 2007 (a), 2009 (b) and 2017(c), whose wind roses at ground-level are shown in Figure 5, winds are stronger on days before PM levels rise, and remain below 3 m s$^{-1}$ when high aerosol concentrations are observed. Each time, the establishment of the high-pressure system brings in air masses from the East (a), or North-East (b and c). Once settled above Central Europe, between 45°N and 50°N, the high largely weakens wind speed (c) or completely nullifies it (a and b).

We note that in the case of January 2017, winds keep a privileged direction according to reanalyses, as the Paris region remains at the edge of the high. Still, those winds from the North-East do not permit any dilution of an aerosol load covering Northern France, Benelux and Western Germany, as shown by ensemble reanalyses of chemical transport models available on the CAMS website (https://atmosphere.copernicus.eu/).

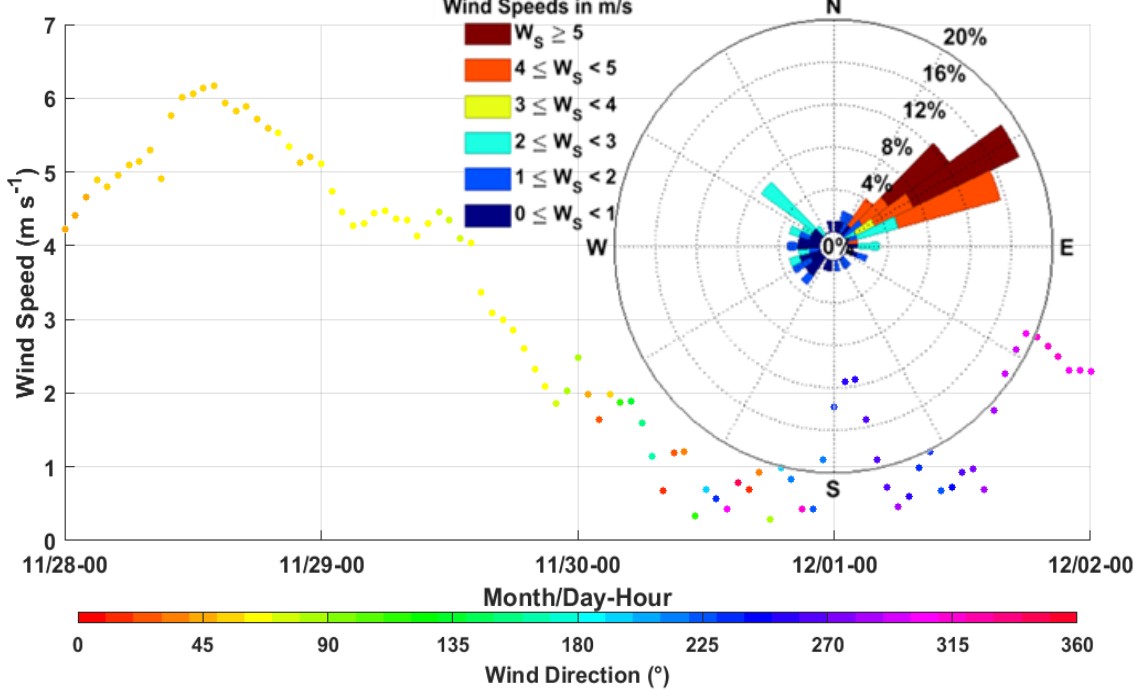

**Figure 4: Temporal evolution of wind intensity and direction derived from ERA5 during four days of the 2016 pollution episode.**





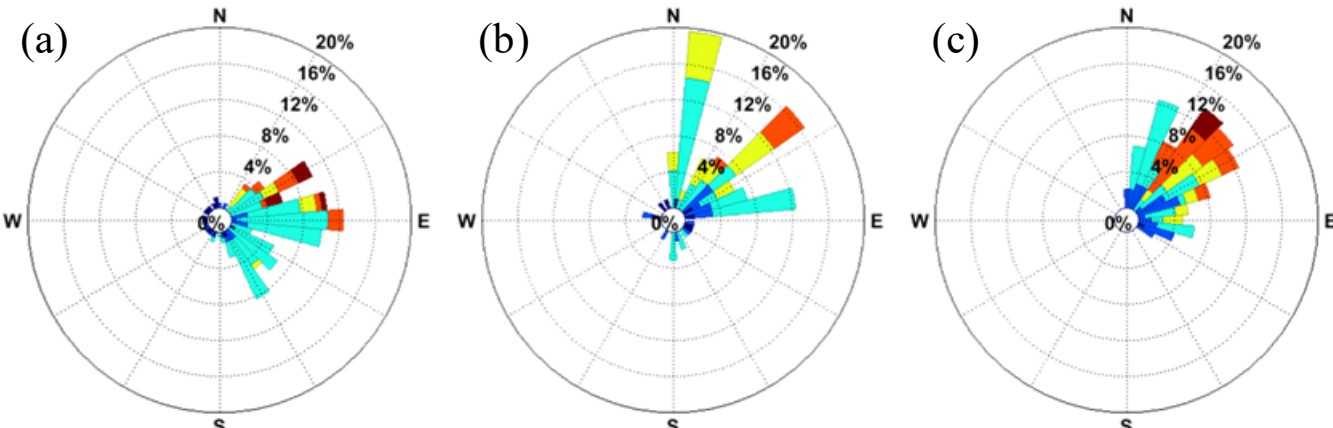

**Figure 5: Wind roses compiling ERA5 data for 7 (a), 4 (b) and 6 (c) days respectively for the 2007, 2009 and 2017 pollution episodes. The colour code is the same as in Figure 4.**

## 4 Lidar-derived aerosol optical properties

### 4.1 AEC and AOT

Figure 6 shows the temporal evolution of the vertical profiles of the AEC during the two pollution events. The AEC is a good proxy for the aerosol load found in the atmospheric column. As shown in Figure 6a, before 30th November, the sky is rather clear and the PBL height reaches approximatively 700 m above ground-level (AGL) with AEC lower than 0.3 km$^{-1}$. From 30th November, the AEC significantly increases with a maximum value close to 0.5 km$^{-1}$. In the evening and the following night, aerosols are trapped within the first 400 m AGL and the AEC reaches values close to 1 km$^{-1}$ at 1200 UTC on 1st December. The PBL remains constrained to 400±50 m AGL until the end of the pollution event on 2nd December.

Figure 6b shows a sharp variation of the PBL top between 21st and 22nd January, where it decreases from 700 to less than 300 m AGL. This decrease is closely linked to the installation of an anticyclonic system over most of Western Europe (see Section 3.2). The AEC remains of the same order of magnitude as during the first pollution event. The two pollution events share a drastic PBL thickness abatement and high AEC values, but the first event appears more suddenly.

The lidar-derived AOT (AOT$_{lid}$) is obtained at 355 nm by integrating the AEC profiles, while the AERONET-derived AOT (AOT$_{phot}$) is computed from AOT$_{phot}$ at 440 nm to 355 nm using the Ångström law (Ångström, 1964). As shown in Figure 7a and Figure 8a, the lidar-derived AOT matches the sun-photometer-derived AOT except on 1st December (Figure 7a) and 21st-22nd January (Figure 8b). These discrepancies are mainly due to the presence of middle and high-altitude clouds identified on lidar vertical profiles, which may bias the AERONET operational products (Chew et al., 2011). For the two winter pollution events of 2016/2017, the AOT at 355 nm remains below 0.5. Note that for 1st December (22nd January), when comparing the MODIS-derived AOT at 550 nm of 0.12±0.07 (0.15±0.07) with the AOT$_{phot}$ of 0.16±0.06 (0.11±0.03) at the same wavelength, they match within 0.04, included in the error bars. All the available values of AOT are summarized in Table 3. For early events in the decade, cloud cover made the availability of AERONET and MODIS level 2 products very rare; only one value from



MODIS on the 22$^{nd}$ December 2007 with AOT = 0.16±0.07 is available. An example of the AOT field as derived from MODIS is given in Figure 9 on 21$^{st}$ January 2017. This highlights the horizontal extent of the pollution plume when observations are not contaminated by clouds. The AOT field appears to be homogeneous within its spatial extension. It is therefore likely that the conclusions deduced from the observations on Paris city may be generalized to a larger spatial scale.

**Figure 6: Temporal evolution of the aerosol extinction coefficient (AEC) at 355 nm as a function of time and altitude for the two cases discussed (a) in late 2016 and b) in January 2017. The colour set from blue to dark red shows an AEC from almost 0 to above 0.6 km⁻¹. White stripes correspond to the presence of clouds.**



## 4.2 Lidar Ratio

The Lidar Ratio varies with the types of aerosols present in the atmospheric column (Müller et al., 2007; Omar et al., 2009; Amiridis et al., 2011; Chazette et al., 2016;). Figure 7a and Figure 8a show the temporal evolution of LR derived from the $N_2$-Raman ground-based lidar for the two APEs of the winter of 2016/2017.

The LR is quite variable for the December APE, with values ranging from ~30 to ~90 sr and a mean value of 59±18 sr. These temporal variations trace a diurnal evolution with smaller aerosol during nighttime, as highlighted on Figure 7b, when the $PM_{2.5}$ to $PM_{10}$ ratio slightly increases during the night (e.g. from 0.4 to 0.6 during the first night, 28[th] November). This increase can be explained by the diurnal variation of aerosol production in an urban area (Airparif, 2014). Indeed, mechanical processes
inducing abrasion (tires, breaks …) linked to human activity decrease during the night. Yet, they are the main source of coarse particles with aerodynamic diameters larger than 2.5 μm. The fine fraction of AOT given by the AERONET operational product is also plotted in Figure 7b (available only during daytime). It agrees with an increase of LR when the load of smaller particle increases from one day to the next. Another explanation would be the diurnal cycle of the relative humidity driven by the diurnal cycle of the temperature. If the aerosols are hygroscopic the diurnal cycle of RH may influence the observed diurnal
cycle of the LR. Randriamiarisoa et al. (2006) have shown that such effect can arise with a RH > 60 % allowing the deliquescence of a hygroscopic aerosol. In our study of winter APEs, the weather conditions are anticyclonic with subsidence of dry air masses. As a result, the RH measured by radiosondes and modelled by ERA5 does not exceed 60% while we observe the diurnal variations of the LR (29[th] November to 1[st] December).

For the APE in January, the LR is pretty much constant over time, ranging between 40 and 50 sr, with a mean value of 45±7
sr (Figure 8a). This period is associated with almost constant values of the fine fraction of aerosol (94±2%) and a ratio of $PM_{2.5}/PM_{10}$ of 77±6 % (Figure 8b).

When only the most polluted days are considered, i.e. 1[st] December and 22[nd] January, the LR increases to 74±16 sr and 56±15 sr, respectively. The presence of smaller particles may be suspected during the first pollution event (sun-photometer-derived visible Ångström exponent of 1.5±0.1 compared to 1.1±0.3 for the second event, see Table 3), likely due to specific
meteorological circulation (see Section 3.2) and the presence of younger aerosols. PDR also corroborate this assumption (Table 3) with higher values in December than in January, whether it is from CALIOP data at 532 nm (9% versus 6%) or the ground-based lidar data at 355 nm (10% versus 5%).

Figure 9 shows that the CALIOP and CATS spaceborne observations may be complementary. Within a 24-hour time interval, their tracks are crossing in the middle of France, along a south-north axis for CALIOP and a west-east axis for CATS. For the
nocturnal orbit on 30[th] November - 1[st] December, the CATS operational product types aerosols as "Polluted continental" corresponding to a LR of 65 sr at 532 nm (see Figure 8a). For the following night, the LR set by CALIOP is 70±25 sr, corresponding to "Polluted continental / smoke" aerosol type, which is coherent with the CATS operational product (Table 3). The LR given by the two spaceborne lidars matches the values derived from the ground-based lidar, although it is not the same





wavelength. Note that Müller et al. (2007) show that the difference on the LR between 355 and 532 nm is in the range of 10% for urban haze aerosols. Our results are consistent with the ones of these authors for urban haze in Central Europe and North America showing lidar-derived LR at 355 nm of 58±12 sr and 53±10 sr, respectively. The aerosol typing derived from CATS at 0300 UTC on 20[th] January and CALIOP at 0200 UTC on 21[st] January is identical to the one previously retrieved for the event of December (Table 3, Figure 7a) although the LR retrieved from the ground-based lidar decreases. Such a discrepancy is not significant when considering the expected uncertainty on the LR given by CALIOP (25 sr).

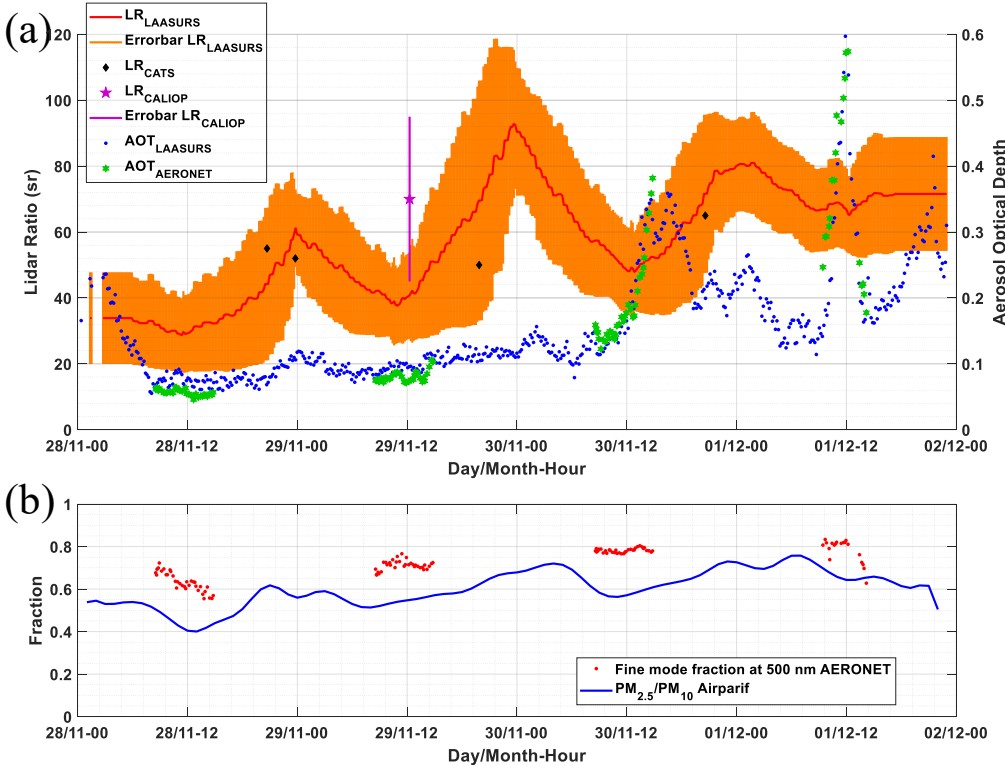

**Figure 7: Time series of Lidar Ratio (LR) and AOT at 355 nm on November-December 2016 (a). On the right Y-axis, the LR as retrieved from lidar measurements is presented in red. The orange area is the associated standard deviation. The LRs extracted from CATS and CALIOP operational products are represented as black diamonds and a purple star, respectively. On the left Y-axis, AOTs at 355 nm retrieved from LAASURS and AERONET are represented in blue and green. The temporal evolution of the fine mode fraction operational product from AERONET is plotted along the $PM_{2.5}$ to $PM_{10}$ ratio from Airparif measurements (b).**


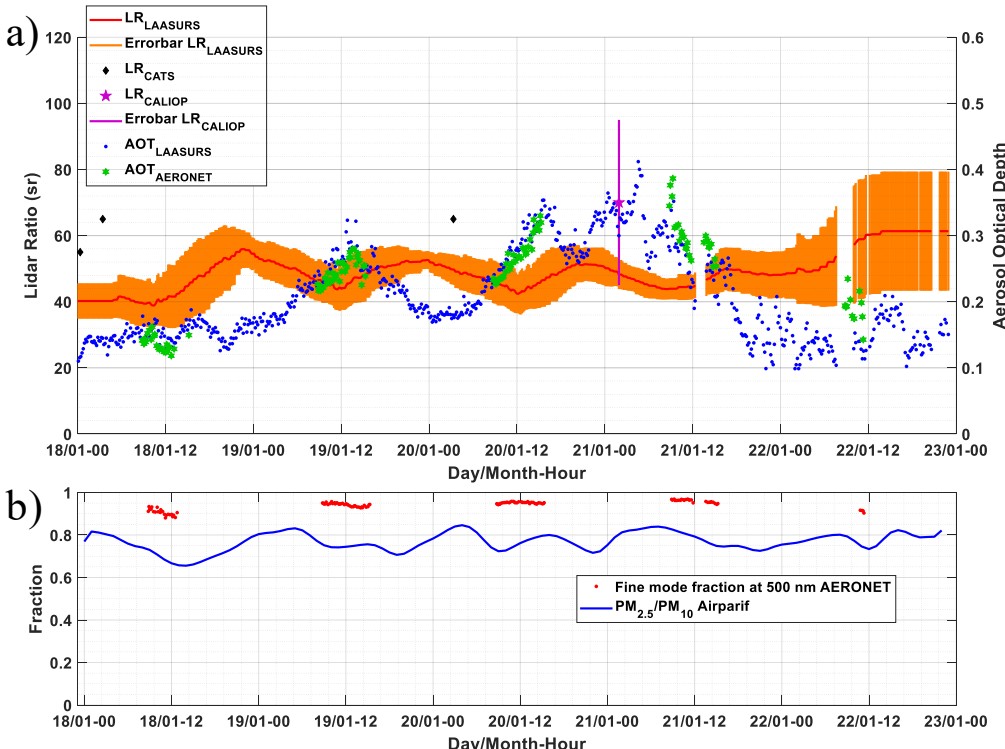

**Figure 8: Same as Figure 7 but for the January 2017 pollution event**



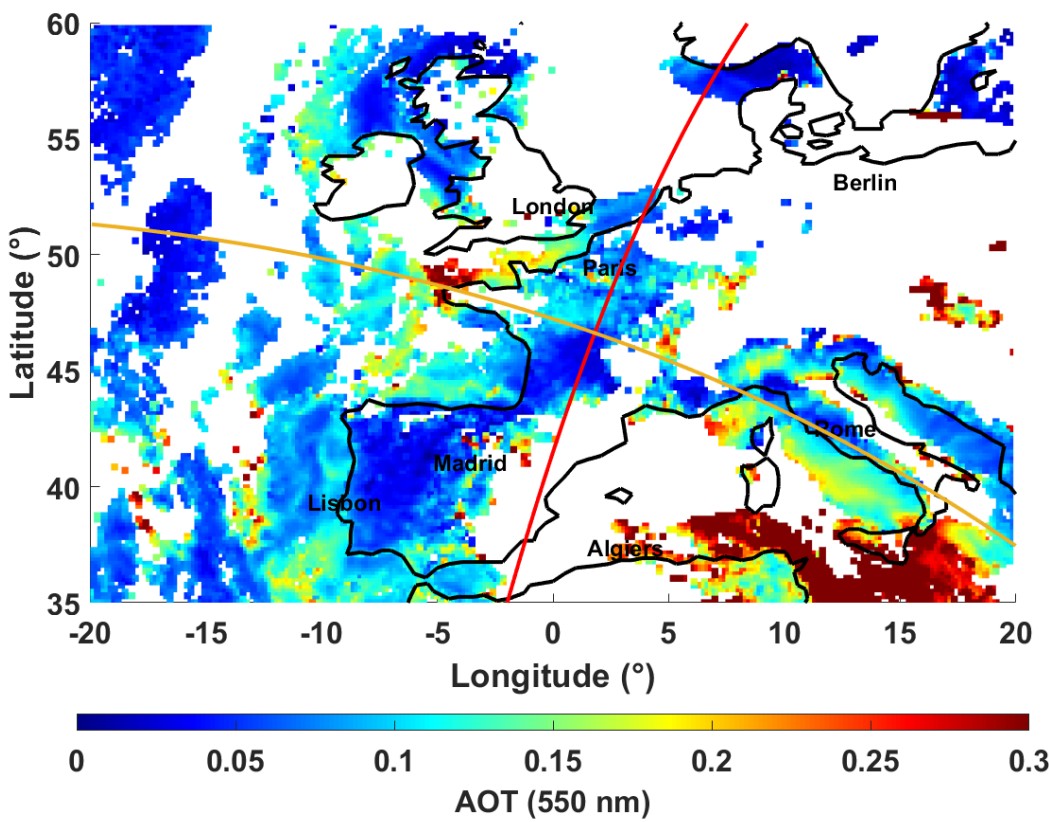

**Figure 9: MODIS-derived aerosol optical thickness (AOT) at 550 nm on 21st January 2017. Red and orange solid lines, respectively show the ground-tracks of CALIPSO/CALIOP (0319 UTC 20th January) and ISS/CATS (0210 UTC 21st January).**

**Table 3: Optical properties encountered during the two most polluted days of the winter 2016/2017.**

| | Date | 2016 APE | 2017 APE |
|---|---|---|---|
| **AOT** | **AERONET** 355 nm / 550 nm | 0.32±0.1 | 0.18±0.02 |
| | | 0.16±0.06 | 0.11±0.03 |
| | **MODIS** 550 nm | 0.12±0.07 | 0.15±0.07 |
| | **LAASURS** 355 nm | 0.23±0.09 | 0.15±0.03 |
| **å** | **AERONET** 340 nm / 675 nm | 1.5±0.1 | 1.1±0.3 |
| **LR (sr)** | **CALIOP** 532 nm | 70±25[1] | 70±25[3] |
| | **CATS** 532 nm | 65[2] | 65[4] |
| | **LAASURS** 355 nm | 72±15 | 56±15 |
| **PDR** | **CALIOP** 532 nm | 0.09[2] | 0.06[3] |
| | **LAASURS** 355 nm | 0.10±0.03 | 0.05±0.02 |



¹ data on 29/11/2016; ² data on 30/11/2016; ³data on 21/01/2017; ⁴data on 20/01/2017

## 5 In situ ground-based measurements versus lidar inversed data

### 5.1 $PM_{10}$ and $PM_{2.5}$ at ground-level

The temporal evolutions of PM during the two particulate pollution events of winter 2016-2017 are analysed. Figure 10 displays both $PM_{2.5}$ and $PM_{10}$ during the APEs of December 2016 and January 2017. Only the background stations (BS) are taken into account. For $PM_{10}$ and $PM_{2.5}$, an hourly average is calculated over all these stations in the Paris vicinity. The related standard deviation surrounds each mean value of PM (coloured area). The information and alert thresholds are also represented. Figure 10a shows a continuous increase of both $PM_{2.5}$ and $PM_{10}$ from 29th November to 2nd December. The information threshold of 50 μg m⁻³ for $PM_{10}$ is exceeded around noon on 30th November. The aerosol mass concentrations overtake the alert threshold of 80 μg m⁻³ for $PM_{10}$ during the night of 30th November – 1st December as the PBL top height decreases. $PM_{10}$ averaged over the Paris region reaches 121 μgm⁻³ on 1st December, just as the lidar records a significant enhancement of the AEC in the entire PBL. Figure 10b shows $PM_{10}$ values around 30 μg m⁻³ during the first days, except at the end of the day on 19th January, when the information threshold is exceeded. A significant decrease of the PBL top height occurred on 21st January at nightfall; preventing the dilution of aerosol, it leads to a strong increase of $PM_{10}$. Indeed, as seen in Figure 6b, the PBL height is divided by two during 21st January and $PM_{10}$ doubles in value during the same time interval. The standard deviation of both $PM_{2.5}$ and $PM_{10}$ are larger on Figure 10a than on Figure 10b. It indicates a greater geographical variability of the pollution plume during the first APE of December. This suggests that PMs are more sensitive to local aerosol sources for the December event than for the one of January.





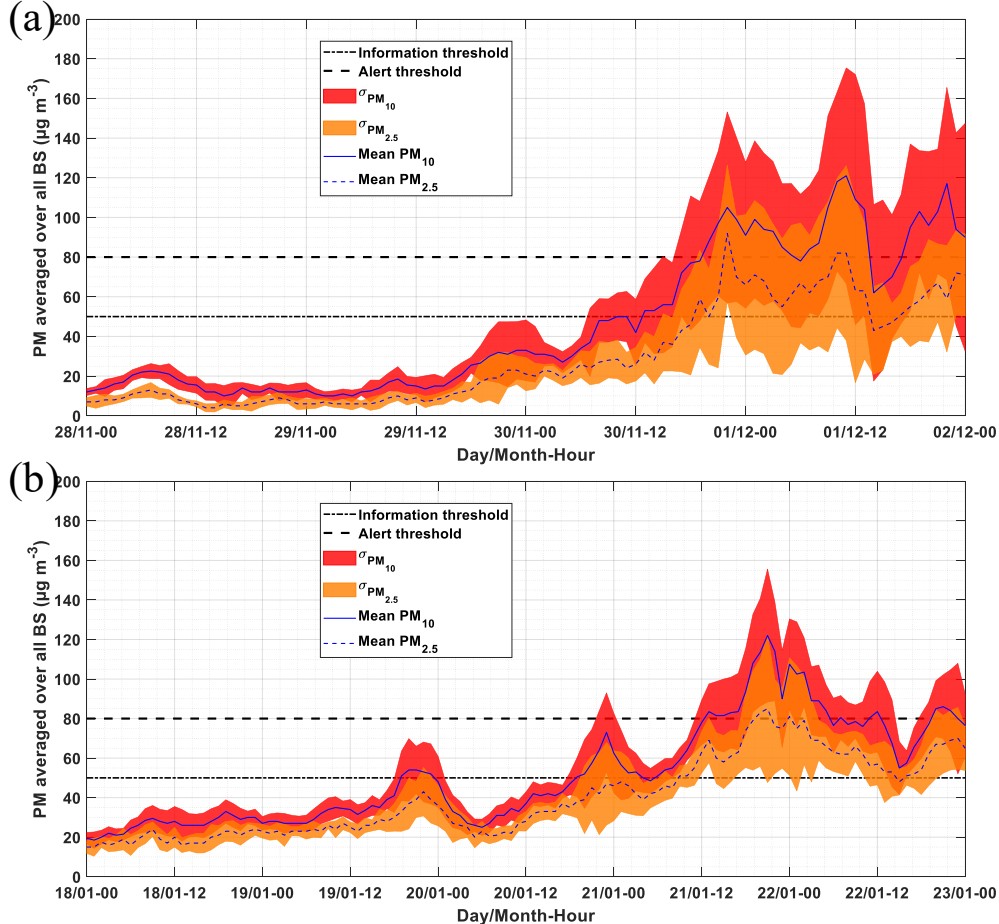

**Figure 10: Temporal evolution of hourly ground PM$_{2.5}$ and PM$_{10}$ during the aerosol pollution events of (a) December 2016 and (b) January 2017. The lines are averages over all the background stations of Ile-de-France, respectively for PM$_{2.5}$ and PM$_{10}$; the coloured areas highlight the standard deviations.**

5 **5.2 Relationship between aerosol optical properties and PM$_{2.5}$**

The parameter legally used to gage an APE is PM$_{10}$ (n° 2008/50/CE and 2004/107/CE), however, Randriamiarisoa et al. (2006) demonstrate that the accumulation mode (PM$_{2.5}$) contributes the most to optical properties of an aerosol population in the Paris area. Thus, in search of a correlation between Figure 6 and Figure 10, we chose PM$_{2.5}$ over PM$_{10}$. We consider the dataset combining the two events displayed in Figure 6, Figure 7 and Figure 10. It ranges from pollution free days to severely polluted

10 days and thus covers a wide range of AEC, AOT and PM$_{2.5}$ values.

Figure 11 shows the scatterplot between the PM$_{2.5}$ measured at ground level and the total AOT$_{lid}$. The linear regression conducted on all the data of Figure 11 (dashed grey line), i.e. AOT and PM$_{2.5}$ day and night for the two pollution episodes sampled by lidar, shows no correlation with a Pearson correlation coefficient R² ≈ 0.16. However, a group of points stands out from this dataset and is associated with a PBL top below 600 m AMSL. It appears that most of these points are also associated





with $PM_{2.5}$ values above the information threshold (50 µg m$^{-3}$). The linear regression conducted without this set of points (black solid line of Figure 11) shows a better correlation ($R^2 \approx 0.66$). It suggests that the direct correlation between AOT and $PM_{2.5}$ to assess air quality, as proposed by Wang and Christopher (2003), Gupta et al. (2006) and Kacenelenbogen (2006), cannot be used under low PBL height conditions. Scatterplots (not shown) made with $PM_{10}$ instead of $PM_{2.5}$ show even worse

correlation ($R^2 \approx 0.03$ and $R^2 \approx 0.61$, respectively).

In Figure 12 the AOT is divided by the boundary layer height derived from the lidar profiles as in Menut et al. (1999), estimating a column-average AEC in the PBL. This technique is used to improve the correlation in Koelemeijer et al. (2006). It assumes that there is no significant contribution of aerosols in the free troposphere; as explained by the authors it is not always the case. We find a significant improvement with the Pearson correlation coefficient rising to ~0.61. This AOT to PBL

height ratio is clearly a better proxy to assess the ground-level aerosol concentration from the AOT.

Another proxy appreciating the intensity of the aerosol load within the PBL would be the maximum of the lidar-derived AEC ($AEC_{max}$) within the PBL. Figure 13 shows the scatterplot of the $AEC_{max}$ against $PM_{2.5}$ for the APEs. We find a significant linear relation with a Pearson correlation coefficient of ~0.75. Here, the average altitude where the $AEC_{max}$ is found is ~300±90 m AMSL (~200±90 m AGL). In a stable PBL with barely any wind shear, $AEC_{max}$ is close to the ground and its variations are

comparable to the ones observed on $PM_{2.5}$. Hence, this optical parameter appears as the most appropriate to monitor the evolution of ground-level winter particulate pollution using ground-based lidar measurements, whether it is heavily polluted or not. Compared with the previous method, the presence of aerosols in the free troposphere does not bias the linear relationship established. However, this approach should not be generalized too quickly for well-developed PBLs that may have high relative humidity at their top. In the case of hydrophilic aerosols, as is often the case for Paris pollution aerosols (Randriamiarisoa et

al., 2006), the $AEC_{max}$ may be near the top of the PBL, here we find 358±229 m as the averaged difference between PBL height and the altitude of $AEC_{max}$ over all the available profiles. Indeed, during winter, at low temperatures, aerosols are generally less acidic and therefore less hydrophilic (Jaffrezo et al., 2005).





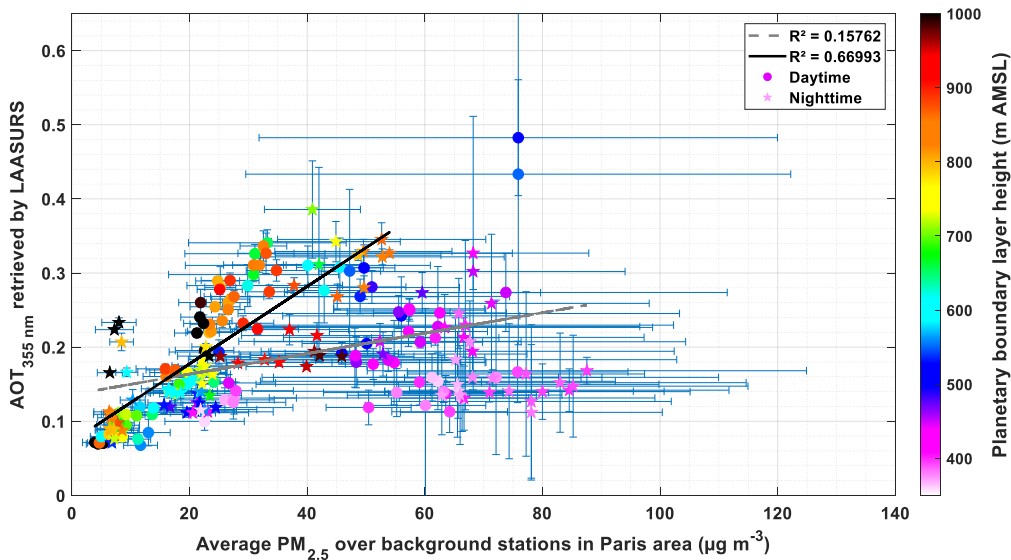

**Figure 11: Relationship between PM$_{2.5}$ (X-axis) and Aerosol Optical Thickness retrieved by lidar at 355 nm (AOT$_{lid}$) (Y-axis) for the dataset presented in Figure 7 and Figure 10 overlapped. The colour set indicates the PBL height retrieved by lidar for each point. Error bars represent the standard deviations due to time average (AOT$_{lid}$) and spatial average (PM$_{2.5}$). The daytime (nighttime) data are represented by discs (stars). The grey dashed line (black solid line) illustrates the linear regression computed from all the trend lines (the set of points associated with a PBL top above 600m AMSL). The correlation coefficients can be found in the top right-hand corner.**

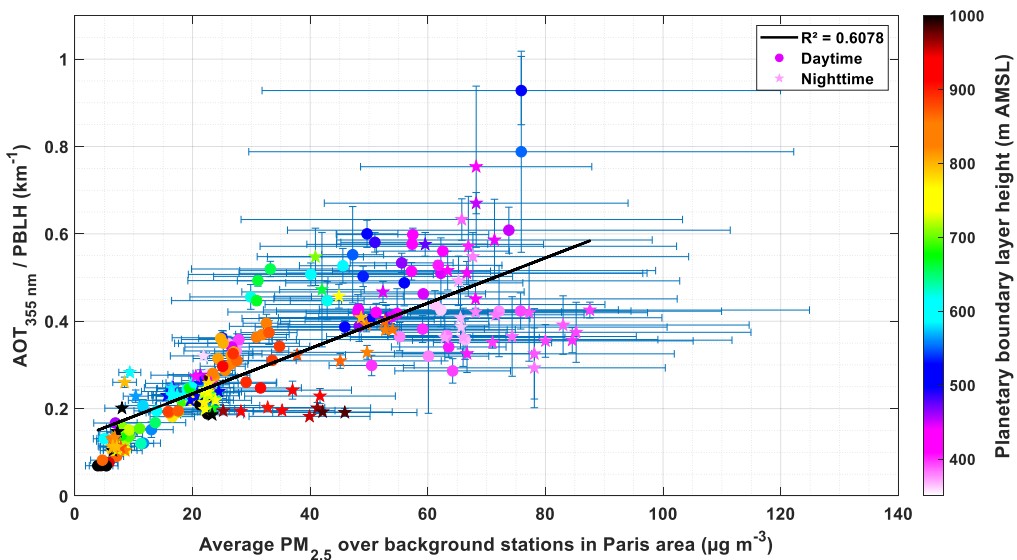

**Figure 12: As Figure 11 but dividing AOT$_{lid}$ by the boundary layer height (BLH) in Y-axis.**





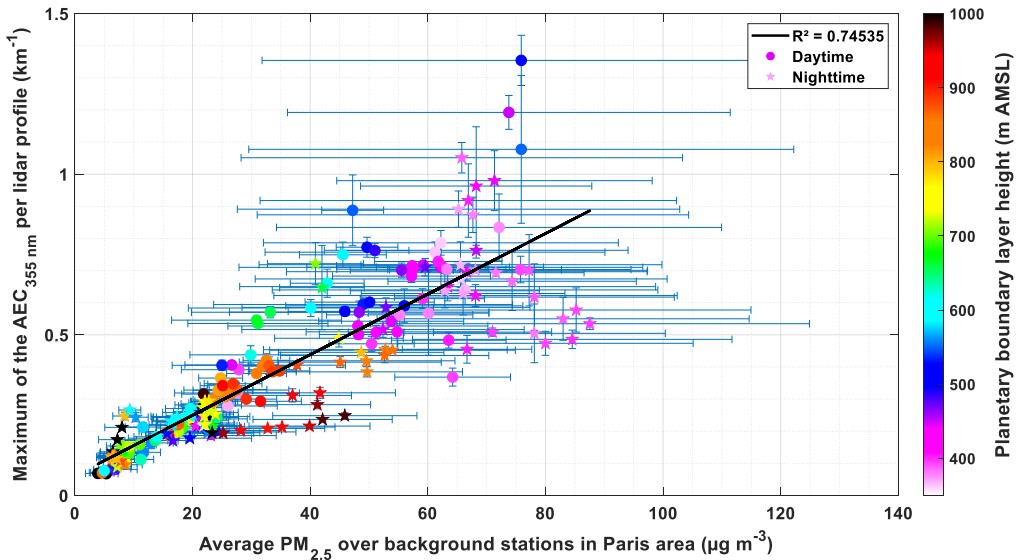

**Figure 13: As Figure 11 but replacing AOT$_{lid}$ by the maximum value of the AEC profile retrieved by the LAASURS within the PBL.**

## 6 Conclusion

In this paper we investigate the lidar-derived optical properties of two major APEs of the winter of 2016/2017, found to be
part of the most severe pollution events of the 2007-2017 decade. The data collected for this study highlights the maximum
AEC in the PBL as an optical parameter that offers the possibility to assess the surface concentration of PM$_{2.5}$. This work is
achieved through a synergy between: i) ground-based active and passive remote sensing devices, ii) spaceborne instruments,
iii) air quality network measurements and iv) meteorological reanalyses.

Although limited in time, this lidar dataset comes to enrich the scientific literature, which was lacking severe winter pollution
data. These episodes are rare (eight days in one decade, split between four separate events), but harmful for the citizens' health
and still difficult to forecast. The two sampled APEs originate from different meteorological processes. The first is triggered
by a high trapping local emissions around a small area which nullifies wind speeds. The other one is provoked by a strong
widespread anticyclone blocking a large area during several days but with advection allowed by remaining winds at its edge.
Furthermore, the suspected presence of younger and finer aerosol in the first APE is corroborated by the higher values of both
LR and PDR retrieved during the aerosol pollution event of December (72±15 sr and 10±3 %, respectively) compared with
the ones of January (56±15 sr and 5±2 %, respectively). In both cases, LR values are confirmed as "polluted continental" by
spaceborne lidars and in accordance with the literature for urban haze.

Our results argue that in stable winter PBL conditions, no linear relationship exists between AOT and particle matter
concentration at ground-level (R²~0.16), i.e. a strong PM$_{2.5}$ at ground-level does not imply a significant AOT within the
atmospheric column. This work shows a better agreement (R²~0.61) when it comes to correlating the surface aerosol



concentration with the PBL-averaged aerosol extinction (AOT to PBL top height ratio) and even better ($R^2 \sim 0.75$) with the maximum of aerosol extinction encountered within the PBL. The latter parameter shows a promising capability to monitor an APE during winter time, as it would not be affected by the aerosol presence above the PBL. A spaceborne active/passive coupling as presented in Toth et al. (2014) with CALIOP and MODIS is limited by clouds present above continental surfaces
nearly 60% of the time (Berthier et al., 2008). However, with a given revisit time of satellites, spaceborne lidar aerosol products could transpose our approach from regional to global scale. Future spaceborne lidar missions such as the ADM-Aeolus (Flamant et al., 2008) and the EarthCARE (Illingworth et al., 2015) satellites could expand the assessment of surface air pollution to a global scale.

**Data availability.** Data can be downloaded upon request from the first author of the paper.

**Author contributions.** Alexandre Baron wrote the paper et analysed the data. Patrick Chazette coordinated and performed the experiment, participated in the analysis and paper writing; Julien Totems performed the experiment and participated in the paper editing.

**Competing interests.** The authors declare that they have no conflict of interest.

**Acknowledgements.** This work was supported by the Commissariat à l'Energie Atomique et aux énergies alternatives (CEA). The Centre National d'Etude Spatial (CNES) helped maintain the Raman-lidar instrument. The authors would like to thank
Pascal Genau (CNRS/LATMOS) and Cristelle Cailteau-Fischbach (UPMC/LATMOS) for their support in operating the lidar data and the welcoming on the site of University Pierre et Marie Curie (UPMC). The authors would like to thank the Airparif network for collecting data. The authors would like to thank the AERONET network for sunphotometer products (at https://aeronet.gsfc.nasa.gov/). The authors acknowledge the MODIS Science, Processing and Data Support Teams for producing and providing MODIS data (at https://modis.gsfc.nasa.gov/data/dataprod/), the Atmospheric Science Data Center
(ASDC) at NASA Langley Research Center (LaRC) for the data processing and distribution of CALIPSO products (level 4.20, at https://search.earthdata.nasa.gov/search) and CATS products (level 2, at https://search.earthdata.nasa.gov/search). ECMWF data used in this study have been obtained from the Copernicus Climate Change Service Climate Data Store (https://cds.climate.copernicus.eu/cdsapp#!/home).

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
