# Peer review of "Remote sensing of two exceptional winter aerosol pollution events and representativeness of ground-based measurements"

_Atmospheric Chemistry and Physics, 2019_

## Referee Comment (RC1) · Anonymous Referee #3 · 21 Aug 2019

In this interesting paper, Baron and co-authors describe ground-based lidar sensing and in situ measurements in Paris, during two high pollution events in winter 2016/17. Surface PM10 reached the 121 and 156 ug/m3 levels, respectively, with an AEC of 0.6-1 km-1 in the second event. The optical properties of the aerosols are characterised, and the measurements are complemented using spaceborne instruments (MODIS, CALIPSO and CATS) and ECMWF reanalyses. The event size around Paris is quantified (250 km diameter) and the boundary layer depth is measured (300 m above ground level). The data collected during the two events is incidentally used

to briefly discuss the relationship between remote sensing observations and surface in-situ measurements.

I believe that the data collected are very valuable, but I have a major objection with the scope of the paper: in its present form, it sets the expectation too high. The title specifies that this is a paper about the "representativeness of the surface - column relationship", and this is reinforced in the first two sentences of the abstract: "In this study an optical parameter derived from lidar measurements is found to be relevant to monitor the evolution of near-surface particulate concentrations. This highlights the opportunities offered by future spaceborne lidar missions in air quality assessment on a global scale."

However, this paper is mainly focused on reporting and describing some very interesting (but necessarily limited) observations. Incidentally, it touches on the issue of the relationship between surface and remote sensing observations, but this is far from being its main topic. If it were, it would use a multitude of datasets to address just this point, and it would not spend much time describing these events themselves in detail. The conclusions drawn are valid for the two events considered, and perhaps for similar meteorological conditions (anticyclonic with a shallow boundary layer). A reader may feel disappointed by how little of the paper is about the topic indicated in the title and abstract.

I suggest that prior to considering this paper for publication, therefore, the words quoted above should be removed from the title and the abstract, which would then start with "This work is carried out following a dedicated field campaign in the Paris area (France) during winter 2016-2017", clarifying from the onset what is the main content of the paper. The incidental study on the representativeness is mentioned towards the end of the abstract and this is ok ("During these two events [...] allows us to investigate the representativeness of optical parameters found in the planetary boundary layer to assess surface aerosol concentration."). What is in my opinion should be avoided is to start with a very broad promise, and then not be able to satisfy the reader's curiosity.

[Figure]

In summary, the dataset is too limited to be suitable for a general study on the above-mentioned representativeness (few observations and specific meteorological conditions). This is was already addressed in the "quick review" report, but it seems that I have not persuaded the authors. I suggest that in the present form of the paper, the authors have not presented sufficient evidence to be able to state that the observations are generally representative of pollution events over Paris in the winter and therefore that the results can hint a relationship between surface and columnar properties, but cannot be considered to be general.

I will also try and produce a detailed review, but please consider the above to be the major point to be addressed (in my opinion).

Best regards.

---

## Referee Comment (RC2) · Anonymous Referee #1 · 24 Aug 2019

[referee-annotated manuscript omitted]

---

## Referee Comment (RC3) · Anonymous Referee #3 · 25 Sep 2019

Please refer also to my previous comment on the fact that "surface - column relationship is touched in the paper but not the main topic". I believe that the data collected are very valuable, but my major objection is with the scope of the paper, as explained therein. The paper as it stands raises the reader's expectations a bit too much.

Detailed suggestions follow below:

MAJOR COMMENTS:

1) I suggest to modify the title and abstract as explained in my previous referee comment (21 August). This is the major objection I have to the paper as it stands now.

2) Another statement that I think could be reviewed on the basis of the above is on page 3, lines 6-7.

3) Angstrom exponent, page 4, lines 25-28. A few points should be clarified in my opinion: (a) when you say "constant" do you mean constant with height or with time? (b) is the Angstrom exponent an instantaneous value or a daily average? (c) the last sentence is unclear (what assumption and what has the horizontal advection to do with it?).

4) Spatial averages, page 8, lines 15-16: I suppose that the way the spatial average is done is limits it to the available ground stations, which means a coarse spatial sampling and a limited overall area. Please mention these caveats in the text.

MINOR COMMENTS:

5) Abstract, line 12: replace "continuously" with "during two 5-day periods" as the lidar was not operated continuosly from 1 November to 31 January (see text).

6) Abstract, line 13: delete "submicron" (at this wavelength the lidar is also sensitive to supermicron particles) and add "thought to be" before "mainly" (you have no direct measurement of aerosol type/origin).

7) Abstract, line 15: explain the method used to determine the circular area and measure its diameter.

8) Abstract, line 17: explain what other information you have to say that the event covered all of Western Europe.

9) Abstract, lines 17 and 18: explain where exactly the values of 121 and 156 have been observed. Explain what is the value after the +/- sign (experimental error? variability in time? standard deviation of measurements at different stations?). You give the AEC of the second episode: why not give also the AEC of the first episode?

10) Abstract, line 20: the sentence about weather conditions is vague, I suggest to be more specific and describe which type of weather conditions you are referring to.

11) Page 3, lines 8-9: "the most severe winter APEs above the Paris area": specify over which period of time they are the most severe (e.g. "from year Y to nowadays").

12) Page 4, line 20: "downgraded" –> "integrated"

13) Page 5, line 6: "sources of uncertainties" –> "uncertainties for our lidar system" (it helps to know that Royer et al is not a generic paper but one that details the uncertainties for this specific lidar).

14) Page 8, line 19: specify over which time period the 136 (27) values are valid (is it the 11 years in Figure 2?).

15) Page 8, lines 21-24: the judgment on pollution could be worded differently, relating to the actual data that are shown. "winter 2016/207 stands out with a large number of threshold exceedances", "2015/16 and 2017/18 had few threshold exceedances".

16) Page 8, line 23: I suggest to omit "despite the increasingly coercing political measures to improve air quality".

17) Page 8, line 24: expand better on the link between pollution levels and anticyclonic conditions.

18) Page 8, lines 29-31: make dates consistent with dates in the abstract, please.

19) Page 9, line 1: next to meteorological patterns add "see section 3.2"

20) Table 2, caption: add "in winter" after "decade"; explain if the max/min value is istantaneous, hourly, daily, etc. In the table, I would suggest to group the event by winter and not year (e.g. 2007/2008 instead of 2007): this would be coherent with Fig. 2.

21) Figure 3, caption: wind velocity and direction at which altitude level? surface?

22) Page 12, line 7: add "single" before "grid point" and give lat/lon of the grid point centre.

23) Fig. 4, x-axis: Month-Day-Hours is confusing; I suggest Day/Month HH:MM. In the caption you should also mention the wind rose.

24) Page 13, line 8: "clear" –> "aerosol-free".

25) Page 13, line 19: state in the paper that you have chosen to keep the data associated with the middle and high altitude clouds and why. Even better, they could be displayed in a different colour for easy identification.

26) Page 15, line 7: are you referring to particle size? I suggest to specify "particle size" after "smaller aerosol"

27) Page 15, line 18: if I have understood your reasoning, then the following sentence could be added for more clarity at the end of this line: "We therefore do not believe that the influence of RH on LR is significant".

28) Page 15, line 28: "shows that the CALIOP and CATS spaceborne observations may be complementary" –> "shows the CALIOP and CATS tracks".

29) Page 19, line 3: add "surface" before "PM".

30) Page 23, line 10: add "over Paris" after "decade".

---

## Author Comment (AC1) · 22 Oct 2019

*Remote sensing of exceptional winter aerosol pollution events and representativeness of the surface – column relationship over Paris metropolitan area*

*by A. Baron, P. Chazette and J. Totems*

In the following, *reviewers' comments are in italic blue*. Responses are in normal black font. **Changes in the text are in black bold**. The numbers of the most major comments are highlighted in red.

**Response to Anonymous Referee #1 – RC2**

We are very grateful to Referee #1 to have reviewed the manuscript and submitted helpful comments and suggestions to improve both the study and the text. Here we respond to the reviewer point by point.

    1) P.1 L.19

*Planetary* boundary layer height *(PBL)*. Changes have been made in the text:

"These two major aerosol pollution events share very low **planetary** boundary layer **(PBL)** heights"

    2) P.1 L.20

*Under the same weather condition*. Changes have been made in the text:

"However, they did not take place **under identical anticyclonic** weather condition**s**"

    3) P.1 L.21

*Aerosol lidar ratio*. Changes have been made in the text:

"they are associated with significantly different **aerosol** lidar ratios**"**

    4) P.2 L.3

*Anthropogenic*. Changes have been made in the text:

"**Anthropogenic** contributors"

    5) P.2 L.9

*Extend*. Changes have been made in the text:

"**extend** from regional to global scale"

    6) P.2 L.21

*Concerned*. Changes have been made in the text:

"is often **concerned** by air pollution issues."

    7) P.2 L.27

. Changes have been made in the text:

"mainly during summer and …"

8) P.2 L.31

*Top*. Changes have been made in the text:

"the PBL height …"

9) P.3 L.1

The use of *in situ* sounding, or *lidar remote sensing techniques to obtain increased vertical and temporal resolution* can *provide valuable data to air pollution models*.

Changes have been made in the text:

"The use of *in situ* sounding, or **lidar remote sensing techniques to obtain increased vertical and temporal resolution** can **provide valuable data to air pollution models**."

10) P.3 L.7

*Planetary boundary layer (PBL)* PBL […] *dinitrogen*. Changes have been made in the text:

"the winter **PBL**." The end of the sentence has been removed due to a significant modification of the sentence asked by the third referee.

11) P.3 L.14

*Such as*. Changes have been made in the text:

"aerosol optical properties, **such as** the lidar ratio"

12) P.3 L.15

*Depolarization ratio (LPDR)*. Changes have been made in the text:

"the linear particle depolari**z**ation ratio (**L**PDR)"

13) P.3 L.16

The meteorological situation *is examined to assign the* origin of pollution. Changes have been made in the text:

"the meteorological situation **is examined to assign the** origin of pollution"

14) P.3 L.18

*Links*. Changes have been made in the text.

"the link between"

15) P.3 L.20

*Instrumentation*. Changes have been made in the text:

"Section 2 presents the **instrumentation** and datasets"

16) P.4 L.31

*Lidar ratio (LR)*. Changes have been made in the text:

"also called **lidar ratio (LR)**."

17) P.5 L.4

*LPDR*. Changes have been made in the text:

"(**L**PDR)"

18) P.5 L.8

SNR is defined P.4 L.19: "To obtain a sufficient signal to noise ratio (SNR)"

19) P.5 L.13

*LPDR*. Changes have been made in the text:

"(**L**PDR)"

20) P.7 L.3

*CATS has been recently evaluated by ground-based lidar measurements from the EARLINET network (Proestakis et al., 2019)*. The sentence and reference have been added.

21) P.10 L.11

*Low-pressure system*. Changes have been made in the text:

"A low**-pressure system** coming from the South"

22) P.10 L.15

The Paris *area*. All the occurrences of "the Paris Area" have been changed in "the Paris **a**rea".

23) P.15 L.2

*It is well established that the LR*. Changes have been made in the text:

"**It is well established that** the LR varies with the types of aerosols"

24) P.15 L.10

"Indeed, mechanical processes inducing abrasion (tires, breaks …) linked to human activity decrease during the night. Yet, they are the main source of coarse particles with aerodynamic diameters larger than 2.5 μm."

*Comment: How the authors are sure about these statements?? Other PM local sources are also present. We know that in Paris area garbage burning industries are located, emitting coarse particles >PM2.5. Could this increase also linked to external sources (advected PMs)? The authors should investigate and comment on these issues.*

It is true that we cannot be affirmative on this particular aspect. To us, the underlying cause of the diurnal variations of both the LR and the $PM_{2.5}/PM_{10}$ ratio could be the resuspension processes more active during daytime. In particular, resuspension of aerosols was identified as a possible cause of discrepancies between models and daytime observations during ESQUIF project (Hodzic et al., 2006).

However, to our knowledge garbage burning industries are no longer coarse particles emitters in the Paris surroundings. Moreover, external sources (e.g. advected particles) are excluded due to the weather and atmospheric structuration conditions (anticyclonic situation, weak winds, and free troposphere and mixing layer un-correlated).

Some changes have been added in the text:

"The LR is quite variable for the December APE, with values ranging from ~30 to ~90 sr and a mean value of 59±18 sr. These temporal variations trace a diurnal evolution with smaller aerosol during night time, as highlighted on **Erreur ! Source du renvoi introuvable.**b, when the $PM_{2.5}$ to $PM_{10}$ ratio slightly increases during the night (e.g. from 0.4 to 0.6 during the first night, 28$^{th}$ November). This increase **may** be explained by the diurnal variation of aerosol production in an urban area (Airparif, 2014). Indeed, mechanical processes inducing abrasion (tires, breaks …) linked to human activity **and resuspensions processes decrease during the night. Yet, they are the main source of coarse particles with aerodynamic diameters larger than 2.5 µm. Resuspension during daytime was highlighted as a possible cause of discrepancies between model and observation during the ESQUIF project in the Paris surroundings (Hodzic et al., 2004, 2006). This could be the underlying cause of the diurnal variations of both the LR and the $PM_{2.5}$ to $PM_{10}$ ratio**. The fine fraction of AOT given by the AERONET operational product is also plotted in **Erreur ! Source du renvoi introuvable.**b (available only during daytime). It agrees with an increase of LR when the load of smaller particle increases from one day to the next."

25) P.15 L.21

*77±6 %*. Changes have been made in the text:

"a ratio of $PM_{2.5}/PM_{10}$ of 77±6 (Figure 8b)."

26) P.16 L.4

*02:00*. In all the paper, the time hours have been written in the format hh:mm.

27) P.16 L.10

*I am not convinced that the CATS and CALIOP trajectories south of Paris area could be compared with data inside the Paris area, as the distance between these 2 sites is very big (>150 km).*

Indeed, the distances between the ground-based lidar and the spaceborne lidars ground-tracks are substantial, ~200km for CATS, the farthest track.

However, according to the figure 9, we see in light blue with an AOT > 0.1 that the pollution plume is quite widespread south of Paris. Each track passes through this plume. According to ensemble reanalyses of chemical transport models available on the CAMS website (https://atmosphere.copernicus.eu/) and given the meteorological conditions discussed above the pollution plume seems to originate from the spreading of the urban haze.

Thus, we assume that the three lidars measured the same type of aerosol and that their data can be compared.

The sentence has been added in the text:

"Erreur ! Source du renvoi introuvable. **shows the CALIOP and CATS ground-tracks for the January APE. Within a 24-hour time interval, their tracks are crossing in the middle of France, along a south-north axis for CALIOP and a west-east axis for CATS. The distances between the ground-based lidar and the spaceborne lidars ground-tracks are substantial (~200km for CATS, the farthest track). However, according to ensemble reanalyses of chemical transport models available on the CAMS website (https://atmosphere.copernicus.eu/) and given the meteorological conditions discussed**

**above, the pollution plume seen by MODIS (AOT > 0.1 in light blue South of Paris) seems to originate from the spreading of the urban haze. The distance separating the ground-based lidar and the farthest ground-track is inferior to the characteristic size of the dispersed plume. Thus, we assume that the spaceborne and ground-based lidars measured the same type of aerosol and that their data are comparable.**"

    28) P.21 L.6

*PBL*. Changes have been made in the text:

"the **PBL** height"

    29) P.21 L.20

*Found*. Changes have been made in the text:

"may be **found** near the top of the PBL"

    30) P.21 L.20

"here we find 358±229 m as the averaged difference between PBL height and the altitude of AEC$_{max}$ over all the available profiles"

*Please comment on this very high value of the std.*

This high value of the standard deviation of the distance between the AEC$_{max}$ altitude and the PBL top can be explain by the wide range of variation of the PBL height in the dataset considered. Indeed, the AEC$_{max}$ altitude is rather stable (~300±90 m AMSL, specified p.21 l.13) but the PBL height is not (~640±250 m AMSL).

Changes in the text:

"In the case of hydrophilic aerosols, as is often the case for Paris pollution aerosols (Randriamiarisoa et al., 2006), the AECmax may be found near the top of the PBL. **Nevertheless, during winter, at low temperatures, aerosols are generally less acidic and therefore less hydrophilic (Jaffrezo et al., 2005).** Here we find 358±229 m as the averaged difference between PBL height and the altitude of AECmax over all the available profiles. **This mean distance is associated with a high standard deviation resulting from the high variation of the PBL height within the considered dataset (~640±250 m AMSL).**"

    31) P.23 L.4

*Investigated*. Changes have been made in the text:

"In this paper we investigate**d** the lidar-derived optical properties"

    32) P.23 L.15

*LPDR*. Changes have been made in the text:

"(**L**PDR)"

---

## Author Comment (AC2) · 22 Oct 2019

*Remote sensing of exceptional winter aerosol pollution events and representativeness of the surface – column relationship over Paris metropolitan area*

*by A. Baron, P. Chazette and J. Totems*

In the following, *reviewers' comments are in italic blue*. Responses are in normal black font. **Changes in the text are in black bold**. The numbers of the most major comments are highlighted in red.

**Response to Anonymous Referee #3 – RC1 & RC3**

We are very grateful to Referee #3 to have reviewed the manuscript and submitted helpful comments and suggestions to improve both the study and the text. Here we respond to the reviewer point by point.

RC1 -

*However, this paper is mainly focused on reporting and describing some very interesting (but necessarily limited) observations. Incidentally, it touches on the issue of the relationship between surface and remote sensing observations, but this is far from being its main topic. If it were, it would use a multitude of datasets to address just this point, and it would not spend much time describing these events themselves in detail. The conclusions drawn are valid for the two events considered, and perhaps for similar meteorological conditions (anticyclonic with a shallow boundary layer). A reader may feel disappointed by how little of the paper is about the topic indicated in the title and abstract.*

*I suggest that prior to considering this paper for publication, therefore, the words quoted above should be removed from the title and the abstract, which would then start with "This work is carried out following a dedicated field campaign in the Paris area (France) during winter 2016-2017", clarifying from the onset what is the main content of the paper. The incidental study on the representativeness is mentioned towards the end of the abstract and this is ok ("During these two events [...] allows us to investigate the representativeness of optical parameters found in the planetary boundary layer to assess surface aerosol concentration."). What is in my opinion should be avoided is to start with a very broad promise, and then not be able to satisfy the reader's curiosity.*

*In summary, the dataset is too limited to be suitable for a general study on the abovementioned representativeness (few observations and specific meteorological conditions). This is was already addressed in the "quick review" report, but it seems that I have not persuaded the authors. I suggest that in the present form of the paper, the authors have not presented sufficient evidence to be able to state that the observations are generally representative of pollution events over Paris in the winter and therefore that the results can hint a relationship between surface and columnar properties, but cannot be considered to be general. I will also try and produce a detailed review, but please consider the above to be the major point to be addressed (in my opinion).*

*Best regards.*

During this first stage of the review process, our manuscript has been quite widely revised following your comments, without losing our scientific objective, mainly dedicated to the interest of lidar measurements to assess the impact of winter particulate pollution.

Still, you consider that the relationship between surface and remote sensing observations is not sufficiently addressed to be the topic of our study. Your comments were overwhelmingly constructive and helpful, yet we partly disagree with you in this last comment. Regarding the finite nature of the

dataset used in the paper, we show in Section 3 that even though the measurements are sampled in a given period of time the dataset are representative of pollution events occurring over the Paris area during winter.

It is difficult to do without the detailed description of the case studies used to establish the relationship between the surface and the PBL. It is indeed important to define them well in order to correctly set the boundary conditions of the study, and this last point answers your objection. Moreover, as we highlight, the two observed events of particulate pollution find equivalents in previous years, which shows that the statistical approach presented in our article is robust and may be generalized to other similar situations that are the majority in winter in the Paris region.

To follow your advice, be more concise and show that our study is mainly based on two major winter pollution events, we have revised the title and the abstract.

*RC3 -

*Please refer also to my previous comment on the fact that "surface - column relationship is touched in the paper but not the main topic". I believe that the data collected are very valuable, but my major objection is with the scope of the paper, as explained therein. The paper as it stands raises the reader's expectations a bit too much.*

*Detailed suggestions follow below:*

➢ MAJOR COMMENTS:

1) *I suggest to modify the title and abstract as explained in my previous referee comment (21 August). This is the major objection I have to the paper as it stands now.*

The title of the paper has been shortened and modified to ensure that there is no ambiguity as to the scope of its results:

"Remote sensing of exceptional winter aerosol pollution events and representativeness **of ground-based measurements**"

The two first sentence of the abstract have also been removed.

2) *Another statement that I think could be reviewed on the basis of the above is on page 3, lines 6-7.*

Changes have been made in the text:

"Hence, the main purpose of this paper is to **describe the meteorological conditions that underlie the establishment of significant winter APEs, characterize observed APEs using in situ and remote sensing data and finally** investigate the link between ground-based aerosol measurements and particles trapped within the winter PBL."

3) *Angstrom exponent, page 4, lines 25-28. A few points should be clarified in my opinion: (a) when you say "constant" do you mean constant with height or with time? (b) is the Angstrom exponent an instantaneous value or a daily average? (c) the last sentence is unclear (what assumption and what has the horizontal advection to do with it?).*

(a) Here, "constant" means constant with height, the sentence has been modified in this way:

(b) The value of Ångström exponent taken is a mean value over the studied period, e.g. a mean value of Å from the 20[th] to the 23[rd] January. Indeed, during such event this value does not present significant

variations. Moreover, E.Dieudonnée et al (2017, referenced in the paper) show that the constant angstrom hypothesis induces a maximum relative uncertainty of 4% in the determination of the LR.

(c) The assumption of a constant Ångström exponent is obsolete in presence of different types of aerosols in the atmospheric column. Indeed, assuming a constant Å in that case would result in an enhanced error on the $N_2$-Raman derived AOT. Thus, we have to argue that we met these conditions under the meteorological context that occurred at the time. See modification

The text has been modified following the three points discussed: (p.4 l.24 to 29)

"**In the inversion process, the extrapolation of AOT measurements from the $N_2$-Raman wavelength to the elastic wavelength assumes a constant Ångström exponent with height for the particles in the atmospheric column. The Ångström exponent is derived from the AErosol RObotic NETwork database (AERONET, [http://aeronet.gsfc.nasa.gov/](http://aeronet.gsfc.nasa.gov/)) for the Paris site. Here a mean value of the Ångström exponent is taken for each APE. This assumption of constant exponent is consistent as all aerosols are concentrated in a well-mixed shallow PBL**."

[...] (p.5 l.10 to 11)

"**Note that Dieudonné et al. (2017) show that the maximum relative uncertainty associated to LR induced by the constant Ångström hypothesis remains below 4%.**"

4) *Spatial averages, page 8, lines 15-16: I suppose that the way the spatial average is done is limits it to the available ground stations, which means a coarse spatial sampling and a limited overall area. Please mention these caveats in the text.*

A sentence has been added following the paragraph discussed to mention this caveat:

"**Even though the network of ground-based stations is designed to be the most representative of the regional air quality, the spatial resolution remains coarse and the average could not be representative of all areas of the Paris region.**"

➤ MINOR COMMENTS:

5) *Abstract, line 12: replace "continuously" with "during two 5-day periods" as the lidar was not operated continuosly from 1 November to 31 January (see text).*

The sentence has been removed because it was not necessary in the abstract.

6) *Abstract, line 13: delete "submicron" (at this wavelength the lidar is also sensitive to supermicron particles) and add "thought to be" before "mainly" (you have no direct measurement of aerosol type/origin).*

The sentence has been removed because it was not necessary in the abstract.

7) *Abstract, line 15: explain the method used to determine the circular area and measure its diameter.*

We used maps of ensemble reanalysis of chemical transport model to assess the dispersion of the APE. In the first case it shows a concentrated zone around the Paris region and its surroundings. In the second event of January the APE is spread in England, Northern France, Benelux and Germany. Such maps are available at [https://atmosphere.copernicus.eu/](https://atmosphere.copernicus.eu/).

Changes have been made in the text:

"it concerned a circular area of ~250 km in diameter around Paris **as shown by ensemble reanalyses of chemical transport models**"

> 8) *Abstract, line 17: explain what other information you have to say that the event covered all of Western Europe.*

Same response as for the point 7).

> 9) *Abstract, lines 17 and 18: explain where exactly the values of 121 and 156 have been observed. Explain what is the value after the +/- sign (experimental error? Variability in time? standard deviation of measurements at different stations?). You give the AEC of the second episode: why not give also the AEC of the first episode?*

The +/- sign is related to standard deviation derived from the spatial averaged.

Changes have been made in the text:

"**The maximum** $PM_{10}$ ($PM_x$ is the mass concentration of particles with an aerodynamic diameter smaller than x μm) **was** 121±63μg m$^{-3}$ **(spatial average ± standard deviation) and the aerosol extinction coefficient (AEC) ranged from 0.2 to 1 km$^{-1}$.** The second event took place from **20$^{th}$ to 23$^{rd}$** January which covered all of **North-western Europe**, with maxima of $PM_{10}$ **around** 156±33 μg m$^{-3}$ and AEC between 0.6 and 1 km$^{-1}$, within the winter atmospheric boundary layer."

> 10) *Abstract, line 20: the sentence about weather conditions is vague, I suggest to be more specific and describe which type of weather conditions you are referring to.*

Changes have been made in the text:

"However, they did not take place **under identical anticyclonic** weather condition**s**"

> 11) *Page 3, lines 8-9: "the most severe winter APEs above the Paris area": specify over which period of time they are the most severe (e.g. "from year Y to nowadays").*

Changes have been made in the text:

"This study is based on a specific field campaign performed during the most severe winter APEs **that occurred** in the Paris area **since 2009**."

> 12) *Page 4, line 20: "downgraded" –> "integrated"*

Agreed. Changes have been made in the text:

"To obtain a sufficient signal to noise ratio (SNR, **SNR > 10 (Royer et al., 2011a)**) from the $N_2$-Raman channel during daytime, the vertical resolution is **set** to 15 m"

> 13) *Page 5, line 6: "sources of uncertainties" –> "uncertainties for our lidar system" (it helps to know that Royer et al is not a generic paper but one that details the uncertainties for this specific lidar).*

Changes have been made in the text:

"The different sources of uncertainties **for our lidar system** are discussed in Royer et al. (2011a)"

> 14) *Page 8, line 19: specify over which time period the 136 (27) values are valid (is it the 11 years in Figure 2?).*

Changes have been made in the text:

"**Among these eleven winters (figure 2),** we count 136 (27) days with at least one station exceeding the information (alert) threshold."

15) *Page 8, lines 21-24: the judgment on pollution could be worded differently, relating to the actual data that are shown. "winter 2016/207 stands out with a large number of threshold exceedances", "2015/16 and 2017/18 had few threshold exceedances".*

Changes have been made in the text:

"Yet, winter 2016/2017 stands out **with a large number of threshold exceedances**. In opposition, the previous and following winters, i.e. 2015/2016 and 2017/2018, **present few threshold exceedances**."

16) *Page 8, line 23: I suggest to omit "despite the increasingly coercing political measures to improve air quality".*

Agreed. The sentence has been removed.

17) *Page 8, line 24: expand better on the link between pollution levels and anticyclonic conditions.*

Three sentences have been added in the text:

"**Indeed, despite a general trend in emissions to decline in the Paris region, there are still noteworthy episodes of pollution. When a strong high pressure system sets in over a long period of time, it prevents air mass advection, blocking the weather situation. Thus, the pollution still emitted, even if it is less than in the past, remains blocked by the high pressure system and ends up exceeding the health thresholds**."

18) *Page 8, lines 29-31: make dates consistent with dates in the abstract, please.*

In the abstract we were referring to the most polluted days of each event. We agree that was not clear, dates in the abstract have been modified.

19) *Page 9, line 1: next to meteorological patterns add "see section 3.2"*

Changes have been made in the text:

"According to ERA5 reanalyses, the meteorological patterns **(see section 3.2)** are similar over the 8 days"

20) *Table 2, caption: add "in winter" after "decade"; explain if the max/min value is istantaneous, hourly, daily, etc. In the table, I would suggest to group the event by winter and not year (e.g. 2007/2008 instead of 2007): this would be coherent with Fig. 2.*

The table header has been changed following your suggestion, and the caption too:

"The 8 most severely polluted days of the past decade **in winter**. For each day we give both $PM_{2.5}$ and $PM_{10}$ measured at ground level (Airparif network) in the format Max/Mean/Min where: Max and Min are the **hourly** maximum and minimum value measured at a given background station during the day and Mean is the daily average over all background stations."

21) *Figure 3, caption: wind velocity and direction at which altitude level? surface?*

Changes have been made in the caption:

"The geopotential altitude **(white lines) and the wind direction and velocity (black arrow) are given** at a 975-hPa level."

22) *Page 12, line 7: add "single" before "grid point" and give lat/lon of the grid point centre.*

Changes have been made in the text:

"We consider the **single** grid point of 0.25° x 0.25° which includes central Paris **(48.875°N, 2.375°E)**."

23) *Fig. 4, x-axis: Month-Day-Hours is confusing; I suggest Day/Month HH:MM. In the caption you should also mention the wind rose.*

The x-axis format has been changed following your suggestion and for more coherence with other figures. Changes have been made in the caption:

"Figure 1: **Wind rose and** temporal evolution of wind intensity and direction **at 10 m** from ERA5 during four days of the 2016 **APE**."

24) *Page 13, line 8: "clear" –> "aerosol-free".*

Changes have been made in the text:

"the sky is rather **aerosol-free**"

25) *Page 13, line 19: state in the paper that you have chosen to keep the data associated with the middle and high altitude clouds and why. Even better, they could be displayed in a different colour for easy identification.*

Changes have been made in the text:

"These discrepancies are mainly due to the presence of middle and high-altitude clouds identified on lidar vertical profiles, which may bias the AERONET operational products (Chew et al., 2011). **As far as lidar data are not disturbed by high clouds these profiles are kept in the figure.**"

26) *Page 15, line 7: are you referring to particle size? I suggest to specify "particle size" after "smaller aerosol".*

Changes have been made in the text:

"These temporal variations trace a diurnal evolution with smaller **particle size** during night time"

27) *Page 15, line 18: if I have understood your reasoning, then the following sentence could be added for more clarity at the end of this line: "We therefore do not believe that the influence of RH on LR is significant".*

Changes have been made in the text:

"As a result, the RH measured by radiosondes and modelled by ERA5 does not exceed 60% while we observe the diurnal variations of the LR (29$^{th}$ November to 1$^{st}$ December). **We therefore do not believe that the influence of RH on LR is significant, as also demonstrated during the LISAIR field campaign (Raut and Chazette, 2007) for RH < 80%.**"

28) *Page 15, line 28: "shows that the CALIOP and CATS spaceborne observations may be complementary" –> "shows the CALIOP and CATS tracks".*

Changes have been made in the text:

"**Erreur ! Source du renvoi introuvable.** shows the CALIOP and CATS **ground-tracks for the January APE**. Within a 24-hour time interval, their tracks are crossing in the middle of France, along a south-north axis for CALIOP and a west-east axis for CATS."

29) *Page 19, line 3: add "surface" before "PM".*

Changes have been made in the text:

"The temporal evolutions of **surface** PM during the two particulate pollution events"

30) *Page 23, line 10: add "over Paris" after "decade".*

Changes have been made in the text:

"In this paper we investigate the lidar-derived optical properties of two major APEs of the winter of 2016/2017, found to be part of the most severe pollution events of the 2007-2017 decade **over Paris**."

---

## Editor Decision (ED1)

*Remote sensing of exceptional winter aerosol pollution events and representativeness of the surface – column relationship over Paris metropolitan area*

*by A. Baron, P. Chazette and J. Totems*

**Response to referees**

Dear Editor,

Please find hereafter the response to the referee's comments. The authors thank the reviewers for thoughtful and constructive insights on the manuscript. We appreciate the time they invested in the review. We believe that our revised manuscript addresses all the comments.

Kind regards,

Alexandre

In the following, *reviewers' comments are in italic blue*. Responses are in normal black font. **Changes in the text are in black bold**. The numbers of the most major comments are highlighted in red.

**Response to Anonymous Referee #1 – RC2**

We are very grateful to Referee #1 to have reviewed the manuscript and submitted helpful comments and suggestions to improve both the study and the text. Here we respond to the reviewer point by point.

1) P.1 L.19

*Planetary* boundary layer height *(PBL)*. Changes have been made in the text:

"These two major aerosol pollution events share very low **planetary** boundary layer **(PBL)** heights"

2) P.1 L.20

*Under the same weather condition*. Changes have been made in the text:

"However, they did not take place **under identical anticyclonic** weather condition**s**"

3) P.1 L.21

*Aerosol lidar ratio*. Changes have been made in the text:

"they are associated with significantly different **aerosol** lidar ratios**"**

4) P.2 L.3

*Anthropogenic*. Changes have been made in the text:

"**Anthropogenic** contributors"

5) P.2 L.9

*Extend*. Changes have been made in the text:

"**extend** from regional to global scale"

    6)  P.2 L.21

*Concerned*. Changes have been made in the text:

"is often **concerned** by air pollution issues."

    7)  P.2 L.27

. Changes have been made in the text:

"mainly during summer and ..."

    8)  P.2 L.31

. Changes have been made in the text:

"the PBL height ..."

    9)  P.3 L.1

The use of *in situ* sounding, or *lidar remote sensing techniques to obtain increased vertical and temporal resolution* can *provide valuable data to air pollution models*.

Changes have been made in the text:

"The use of *in situ* sounding, or **lidar remote sensing techniques to obtain increased vertical and temporal resolution** can **provide valuable data to air pollution models**."

    10) P.3 L.7

 *PBL* [...] . Changes have been made in the text:

"the winter **PBL**." The end of the sentence has been removed due to a significant modification of the sentence asked by the third referee.

    11) P.3 L.14

*Such as*. Changes have been made in the text:

"aerosol optical properties, **such as** the lidar ratio"

    12) P.3 L.15

*Depolarization ratio (LPDR)*. Changes have been made in the text:

"the linear particle depolari**z**ation ratio (**L**PDR)"

    13) P.3 L.16

The meteorological situation *is examined to assign the* origin of pollution. Changes have been made in the text:

"the meteorological situation **is examined to assign the** origin of pollution"

    14) P.3 L.18

*Links*. Changes have been made in the text.

"the link between"

15) P.3 L.20

*Instrumentation*. Changes have been made in the text:

"Section 2 presents the **instrumentation** and datasets"

16) P.4 L.31

*Lidar ratio (LR)*. Changes have been made in the text:

"also called **lidar ratio (LR)**."

17) P.5 L.4

*LPDR*. Changes have been made in the text:

"(**L**PDR)"

18) P.5 L.8

SNR is defined P.4 L.19: "To obtain a sufficient signal to noise ratio (SNR)"

19) P.5 L.13

*LPDR*. Changes have been made in the text:

"(**L**PDR)"

20) P.7 L.3

*CATS has been recently evaluated by ground-based lidar measurements from the EARLINET network (Proestakis et al., 2019)*. The sentence and reference have been added.

21) P.10 L.11

*Low-pressure system*. Changes have been made in the text:

"A low-**pressure system** coming from the South"

22) P.10 L.15

The Paris *area*. All the occurrences of "the Paris Area" have been changed in "the Paris **a**rea".

23) P.15 L.2

*It is well established that the LR*. Changes have been made in the text:

"**It is well established that** the LR varies with the types of aerosols"

24) P.15 L.10

"Indeed, mechanical processes inducing abrasion (tires, breaks …) linked to human activity decrease during the night. Yet, they are the main source of coarse particles with aerodynamic diameters larger than 2.5 μm."

*Comment: How the authors are sure about these statements?? Other PM local sources are also present. We know that in Paris area garbage burning industries are located, emitting coarse particles >PM2.5. Could this increase also linked to external sources (advected PMs)? The authors should investigate and comment on these issues.*

It is true that we cannot be affirmative on this particular aspect. To us, the underlying cause of the diurnal variations of both the LR and the $PM_{2.5}/PM_{10}$ ratio could be the resuspension processes more active during daytime. In particular, resuspension of aerosols was identified as a possible cause of discrepancies between models and daytime observations during ESQUIF project (Hodzic et al., 2006).

However, to our knowledge garbage burning industries are no longer coarse particles emitters in the Paris surroundings. Moreover, external sources (e.g. advected particles) are excluded due to the weather and atmospheric structuration conditions (anticyclonic situation, weak winds, and free troposphere and mixing layer un-correlated).

Some changes have been added in the text:

"The LR is quite variable for the December APE, with values ranging from ~30 to ~90 sr and a mean value of 59±18 sr. These temporal variations trace a diurnal evolution with smaller aerosol during night time, as highlighted on **Erreur ! Source du renvoi introuvable.**b, when the $PM_{2.5}$ to $PM_{10}$ ratio slightly increases during the night (e.g. from 0.4 to 0.6 during the first night, 28[th] November). This increase **may** be explained by the diurnal variation of aerosol production in an urban area (Airparif, 2014). Indeed, mechanical processes inducing abrasion (tires, breaks …) linked to human activity **and resuspensions processes decrease during the night. Yet, they are the main source of coarse particles with aerodynamic diameters larger than 2.5 µm. Resuspension during daytime was highlighted as a possible cause of discrepancies between model and observation during the ESQUIF project in the Paris surroundings (Hodzic et al., 2004, 2006). This could be the underlying cause of the diurnal variations of both the LR and the $PM_{2.5}$ to $PM_{10}$ ratio**. The fine fraction of AOT given by the AERONET operational product is also plotted in **Erreur ! Source du renvoi introuvable.**b (available only during daytime). It agrees with an increase of LR when the load of smaller particle increases from one day to the next."

25) P.15 L.21

*77±6 %*. Changes have been made in the text:

"a ratio of $PM_{2.5}/PM_{10}$ of 77±6 (Figure 8b)."

26) P.16 L.4

*02:00*. In all the paper, the time hours have been written in the format hh:mm.

27) P.16 L.10

*I am not convinced that the CATS and CALIOP trajectories south of Paris area could be compared with data inside the Paris area, as the distance between these 2 sites is very big (>150 km).*

Indeed, the distances between the ground-based lidar and the spaceborne lidars ground-tracks are substantial, ~200km for CATS, the farthest track.

However, according to the figure 9, we see in light blue with an AOT > 0.1 that the pollution plume is quite widespread south of Paris. Each track passes through this plume. According to ensemble

reanalyses of chemical transport models available on the CAMS website (https://atmosphere.copernicus.eu/) and given the meteorological conditions discussed above the pollution plume seems to originate from the spreading of the urban haze.

Thus, we assume that the three lidars measured the same type of aerosol and that their data can be compared.

The sentence has been added in the text:

"Erreur ! Source du renvoi introuvable. **shows the CALIOP and CATS ground-tracks for the January APE. Within a 24-hour time interval, their tracks are crossing in the middle of France, along a south-north axis for CALIOP and a west-east axis for CATS. The distances between the ground-based lidar and the spaceborne lidars ground-tracks are substantial (~200km for CATS, the farthest track). However, according to ensemble reanalyses of chemical transport models available on the CAMS website (https://atmosphere.copernicus.eu/) and given the meteorological conditions discussed above, the pollution plume seen by MODIS (AOT > 0.1 in light blue South of Paris) seems to originate from the spreading of the urban haze. The distance separating the ground-based lidar and the farthest ground-track is inferior to the characteristic size of the dispersed plume. Thus, we assume that the spaceborne and ground-based lidars measured the same type of aerosol and that their data are comparable.**"

28) P.21 L.6

*PBL*. Changes have been made in the text:

"the **PBL** height"

29) P.21 L.20

*Found*. Changes have been made in the text:

"may be **found** near the top of the PBL"

30) P.21 L.20

"here we find 358±229 m as the averaged difference between PBL height and the altitude of AEC$_{max}$ over all the available profiles"

*Please comment on this very high value of the std.*

This high value of the standard deviation of the distance between the AEC$_{max}$ altitude and the PBL top can be explain by the wide range of variation of the PBL height in the dataset considered. Indeed, the AEC$_{max}$ altitude is rather stable (~300±90 m AMSL, specified p.21 l.13) but the PBL height is not (~640±250 m AMSL).

Changes in the text:

"In the case of hydrophilic aerosols, as is often the case for Paris pollution aerosols (Randriamiarisoa et al., 2006), the AECmax may be found near the top of the PBL. **Nevertheless, during winter, at low temperatures, aerosols are generally less acidic and therefore less hydrophilic (Jaffrezo et al., 2005).** Here we find 358±229 m as the averaged difference between PBL height and the altitude of AECmax over all the available profiles. **This mean distance is associated with a high standard deviation resulting from the high variation of the PBL height within the considered dataset (~640±250 m AMSL).**"

31) P.23 L.4

*Investigated*. Changes have been made in the text:

"In this paper we investigate**d** the lidar-derived optical properties"

32) P.23 L.15

*LPDR*. Changes have been made in the text:

"(**L**PDR)"

We are very grateful to Referee #3 to have reviewed the manuscript and submitted helpful comments and suggestions to improve both the study and the text. Here we respond to the reviewer point by point.

RC1 -

*However, this paper is mainly focused on reporting and describing some very interesting (but necessarily limited) observations. Incidentally, it touches on the issue of the relationship between surface and remote sensing observations, but this is far from being its main topic. If it were, it would use a multitude of datasets to address just this point, and it would not spend much time describing these events themselves in detail. The conclusions drawn are valid for the two events considered, and perhaps for similar meteorological conditions (anticyclonic with a shallow boundary layer). A reader may feel disappointed by how little of the paper is about the topic indicated in the title and abstract.*

*I suggest that prior to considering this paper for publication, therefore, the words quoted above should be removed from the title and the abstract, which would then start with "This work is carried out following a dedicated field campaign in the Paris area (France) during winter 2016-2017", clarifying from the onset what is the main content of the paper. The incidental study on the representativeness is mentioned towards the end of the abstract and this is ok ("During these two events [...] allows us to investigate the representativeness of optical parameters found in the planetary boundary layer to assess surface aerosol concentration."). What is in my opinion should be avoided is to start with a very broad promise, and then not be able to satisfy the reader's curiosity.*

*In summary, the dataset is too limited to be suitable for a general study on the abovementioned representativeness (few observations and specific meteorological conditions). This is was already addressed in the "quick review" report, but it seems that I have not persuaded the authors. I suggest that in the present form of the paper, the authors have not presented sufficient evidence to be able to state that the observations are generally representative of pollution events over Paris in the winter and therefore that the results can hint a relationship between surface and columnar properties, but cannot be considered to be general. I will also try and produce a detailed review, but please consider the above to be the major point to be addressed (in my opinion).*

*Best regards.*

During this first stage of the review process, our manuscript has been quite widely revised following your comments, without losing our scientific objective, mainly dedicated to the interest of lidar measurements to assess the impact of winter particulate pollution.

Still, you consider that the relationship between surface and remote sensing observations is not sufficiently addressed to be the topic of our study. Your comments were overwhelmingly constructive and helpful, yet we partly disagree with you in this last comment. Regarding the finite nature of the dataset used in the paper, we show in Section 3 that even though the measurements are sampled in a given period of time the dataset are representative of pollution events occurring over the Paris area during winter.

It is difficult to do without the detailed description of the case studies used to establish the relationship between the surface and the PBL. It is indeed important to define them well in order to correctly set the boundary conditions of the study, and this last point answers your objection. Moreover, as we highlight, the two observed events of particulate pollution find equivalents in previous years, which shows that the statistical approach presented in our article is robust and may be generalized to other similar situations that are the majority in winter in the Paris region.

To follow your advice, be more concise and show that our study is mainly based on two major winter pollution events, we have revised the title and the abstract.

RC3 -

*Please refer also to my previous comment on the fact that "surface - column relationship is touched in the paper but not the main topic". I believe that the data collected are very valuable, but my major objection is with the scope of the paper, as explained therein. The paper as it stands raises the reader's expectations a bit too much.*

*Detailed suggestions follow below:*

➢ MAJOR COMMENTS:
1) *I suggest to modify the title and abstract as explained in my previous referee comment (21 August). This is the major objection I have to the paper as it stands now.*

The title of the paper has been shortened and modified to ensure that there is no ambiguity as to the scope of its results:

"Remote sensing of exceptional winter aerosol pollution events and representativeness **of ground-based measurements**"

The two first sentence of the abstract have also been removed.

2) *Another statement that I think could be reviewed on the basis of the above is on page 3, lines 6-7.*

Changes have been made in the text:

"Hence, the main purpose of this paper is to **describe the meteorological conditions that underlie the establishment of significant winter APEs, characterize observed APEs using in situ and remote sensing data and finally** investigate the link between ground-based aerosol measurements and particles trapped within the winter PBL."

3) *Angstrom exponent, page 4, lines 25-28. A few points should be clarified in my opinion: (a) when you say "constant" do you mean constant with height or with time? (b) is the Angstrom exponent an instantaneous value or a daily average? (c) the last sentence is unclear (what assumption and what has the horizontal advection to do with it?).*

(a) Here, "constant" means constant with height, the sentence has been modified in this way:

(b) The value of Ångström exponent taken is a mean value over the studied period, e.g. a mean value of Å from the 20th to the 23rd January. Indeed, during such event this value does not present significant variations. Moreover, E.Dieudonnée et al (2017, referenced in the paper) show that the constant angstrom hypothesis induces a maximum relative uncertainty of 4% in the determination of the LR.

(c) The assumption of a constant Ångström exponent is obsolete in presence of different types of aerosols in the atmospheric column. Indeed, assuming a constant Å in that case would result in an enhanced error on the $N_2$-Raman derived AOT. Thus, we have to argue that we met these conditions under the meteorological context that occurred at the time. See modification

The text has been modified following the three points discussed: (p.4 l.24 to 29)

"**In the inversion process, the extrapolation of AOT measurements from the $N_2$-Raman wavelength to the elastic wavelength assumes a constant Ångström exponent with height for the particles in the**

**atmospheric column. The Ångström exponent is derived from the AErosol RObotic NETwork database (AERONET, http://aeronet.gsfc.nasa.gov/) for the Paris site. Here a mean value of the Ångström exponent is taken for each APE. This assumption of constant exponent is consistent as all aerosols are concentrated in a well-mixed shallow PBL**."

[...] (p.5 l.10 to 11)

"**Note that Dieudonné et al. (2017) show that the maximum relative uncertainty associated to LR induced by the constant Ångström hypothesis remains below 4%.**"

4) *Spatial averages, page 8, lines 15-16: I suppose that the way the spatial average is done is limits it to the available ground stations, which means a coarse spatial sampling and a limited overall area. Please mention these caveats in the text.*

A sentence has been added following the paragraph discussed to mention this caveat:

"**Even though the network of ground-based stations is designed to be the most representative of the regional air quality, the spatial resolution remains coarse and the average could not be representative of all areas of the Paris region.**"

➤ MINOR COMMENTS:

5) *Abstract, line 12: replace "continuously" with "during two 5-day periods" as the lidar was not operated continuosly from 1 November to 31 January (see text).*

The sentence has been removed because it was not necessary in the abstract.

6) *Abstract, line 13: delete "submicron" (at this wavelength the lidar is also sensitive to supermicron particles) and add "thought to be" before "mainly" (you have no direct measurement of aerosol type/origin).*

The sentence has been removed because it was not necessary in the abstract.

7) *Abstract, line 15: explain the method used to determine the circular area and measure its diameter.*

We used maps of ensemble reanalysis of chemical transport model to assess the dispersion of the APE. In the first case it shows a concentrated zone around the Paris region and its surroundings. In the second event of January the APE is spread in England, Northern France, Benelux and Germany. Such maps are available at https://atmosphere.copernicus.eu/.

Changes have been made in the text:

"it concerned a circular area of ~250 km in diameter around Paris **as shown by ensemble reanalyses of chemical transport models**"

8) *Abstract, line 17: explain what other information you have to say that the event covered all of Western Europe.*

Same response as for the point 7).

9) *Abstract, lines 17 and 18: explain where exactly the values of 121 and 156 have been observed. Explain what is the value after the +/- sign (experimental error? Variability in time? standard deviation*

*of measurements at different stations?). You give the AEC of the second episode: why not give also the AEC of the first episode?*

The +/- sign is related to standard deviation derived from the spatial averaged.

Changes have been made in the text:

"**The maximum** $PM_{10}$ ($PM_x$ is the mass concentration of particles with an aerodynamic diameter smaller than x μm) **was** 121±63μg m$^{-3}$ **(spatial average ± standard deviation) and the aerosol extinction coefficient (AEC) ranged from 0.2 to 1 km$^{-1}$.** The second event took place from **20$^{th}$ to 23$^{rd}$** January which covered all of **North-western Europe**, with maxima of $PM_{10}$ **around** 156±33 μg m$^{-3}$ and AEC between 0.6 and 1 km$^{-1}$, within the winter atmospheric boundary layer."

10) *Abstract, line 20: the sentence about weather conditions is vague, I suggest to be more specific and describe which type of weather conditions you are referring to.*

Changes have been made in the text:

"However, they did not take place **under identical anticyclonic** weather condition**s**"

11) *Page 3, lines 8-9: "the most severe winter APEs above the Paris area": specify over which period of time they are the most severe (e.g. "from year Y to nowadays").*

Changes have been made in the text:

"This study is based on a specific field campaign performed during the most severe winter APEs **that occurred** in the Paris area **since 2009**."

12) *Page 4, line 20: "downgraded" –> "integrated"*

Agreed. Changes have been made in the text:

"To obtain a sufficient signal to noise ratio (SNR, **SNR > 10 (Royer et al., 2011a)**) from the $N_2$-Raman channel during daytime, the vertical resolution is **set** to 15 m"

13) *Page 5, line 6: "sources of uncertainties" –> "uncertainties for our lidar system" (it helps to know that Royer et al is not a generic paper but one that details the uncertainties for this specific lidar).*

Changes have been made in the text:

"The different sources of uncertainties **for our lidar system** are discussed in Royer et al. (2011a)"

14) *Page 8, line 19: specify over which time period the 136 (27) values are valid (is it the 11 years in Figure 2?).*

Changes have been made in the text:

"**Among these eleven winters (figure 2),** we count 136 (27) days with at least one station exceeding the information (alert) threshold."

15) *Page 8, lines 21-24: the judgment on pollution could be worded differently, relating to the actual data that are shown. "winter 2016/207 stands out with a large number of threshold exceedances", "2015/16 and 2017/18 had few threshold exceedances".*

Changes have been made in the text:

"Yet, winter 2016/2017 stands out **with a large number of threshold exceedances**. In opposition, the previous and following winters, i.e. 2015/2016 and 2017/2018, **present few threshold exceedances**."

16) *Page 8, line 23: I suggest to omit "despite the increasingly coercing political measures to improve air quality".*

Agreed. The sentence has been removed.

17) *Page 8, line 24: expand better on the link between pollution levels and anticyclonic conditions.*

Three sentences have been added in the text:

"**Indeed, despite a general trend in emissions to decline in the Paris region, there are still noteworthy episodes of pollution. When a strong high pressure system sets in over a long period of time, it prevents air mass advection, blocking the weather situation. Thus, the pollution still emitted, even if it is less than in the past, remains blocked by the high pressure system and ends up exceeding the health thresholds**."

18) *Page 8, lines 29-31: make dates consistent with dates in the abstract, please.*

In the abstract we were referring to the most polluted days of each event. We agree that was not clear, dates in the abstract have been modified.

19) *Page 9, line 1: next to meteorological patterns add "see section 3.2"*

Changes have been made in the text:

"According to ERA5 reanalyses, the meteorological patterns **(see section 3.2)** are similar over the 8 days"

20) *Table 2, caption: add "in winter" after "decade"; explain if the max/min value is istantaneous, hourly, daily, etc. In the table, I would suggest to group the event by winter and not year (e.g. 2007/2008 instead of 2007): this would be coherent with Fig. 2.*

The table header has been changed following your suggestion, and the caption too:

"The 8 most severely polluted days of the past decade **in winter**. For each day we give both $PM_{2.5}$ and $PM_{10}$ measured at ground level (Airparif network) in the format Max/Mean/Min where: Max and Min are the **hourly** maximum and minimum value measured at a given background station during the day and Mean is the daily average over all background stations."

21) *Figure 3, caption: wind velocity and direction at which altitude level? surface?*

Changes have been made in the caption:

"The geopotential altitude **(white lines) and the wind direction and velocity (black arrow) are given** at a 975-hPa level."

22) *Page 12, line 7: add "single" before "grid point" and give lat/lon of the grid point centre.*

Changes have been made in the text:

"We consider the **single** grid point of 0.25° x 0.25° which includes central Paris **(48.875°N, 2.375°E)**."

23) *Fig. 4, x-axis: Month-Day-Hours is confusing; I suggest Day/Month HH:MM. In the caption you should also mention the wind rose.*

The x-axis format has been changed following your suggestion and for more coherence with other figures. Changes have been made in the caption:

"Figure 1: **Wind rose and** temporal evolution of wind intensity and direction **at 10 m** from ERA5 during four days of the 2016 **APE**."

24) *Page 13, line 8: "clear" –> "aerosol-free".*

Changes have been made in the text:

"the sky is rather **aerosol-free**"

25) *Page 13, line 19: state in the paper that you have chosen to keep the data associated with the middle and high altitude clouds and why. Even better, they could be displayed in a different colour for easy identification.*

Changes have been made in the text:

"These discrepancies are mainly due to the presence of middle and high-altitude clouds identified on lidar vertical profiles, which may bias the AERONET operational products (Chew et al., 2011). **As far as lidar data are not disturbed by high clouds these profiles are kept in the figure.**"

26) *Page 15, line 7: are you referring to particle size? I suggest to specify "particle size" after "smaller aerosol".*

Changes have been made in the text:

"These temporal variations trace a diurnal evolution with smaller **particle size** during night time"

27) *Page 15, line 18: if I have understood your reasoning, then the following sentence could be added for more clarity at the end of this line: "We therefore do not believe that the influence of RH on LR is significant".*

Changes have been made in the text:

"As a result, the RH measured by radiosondes and modelled by ERA5 does not exceed 60% while we observe the diurnal variations of the LR (29$^{th}$ November to 1$^{st}$ December). **We therefore do not believe that the influence of RH on LR is significant, as also demonstrated during the LISAIR field campaign (Raut and Chazette, 2007) for RH < 80%.**"

28) *Page 15, line 28: "shows that the CALIOP and CATS spaceborne observations may be complementary" –> "shows the CALIOP and CATS tracks".*

Changes have been made in the text:

"**Erreur ! Source du renvoi introuvable.** shows the CALIOP and CATS **ground-tracks for the January APE**. Within a 24-hour time interval, their tracks are crossing in the middle of France, along a south-north axis for CALIOP and a west-east axis for CATS."

29) *Page 19, line 3: add "surface" before "PM".*

Changes have been made in the text:

"The temporal evolutions of **surface** PM during the two particulate pollution events"

30) *Page 23, line 10: add "over Paris" after "decade".*

Changes have been made in the text:

[revised manuscript text omitted]

---

## Author Response (AR2)

*Remote sensing of two exceptional winter aerosol pollution events and representativeness of ground-based measurements*

*by A. Baron, P. Chazette and J. Totems*

**Authors response to the referee #5 and Editor**

5 Dear Editor,

Please find hereafter the response to your comments and notes. The authors thank you for thoughtful and constructive insights on the manuscript. We appreciate the time you invested in the review. We believe that our revised manuscript addresses all the comments including the addition of a new dataset with which we test the relationship of Figure 13.

Kind regards,

10 Alexandre

In the following, *comments are in italic blue*. Responses are in normal black font. **Changes in the text are in black bold**. The numbers of the most major comments are highlighted in red.

**Response to the referee #5 comments**

15 *The authors use a combination of ground-based PM measurements and active and passive remote sensing from ground and space to investigate aerosol pollution events in the Paris region. While the general topic is of importance, the present paper does not provide a substantial contribution to scientific progress. The authors present a combination of the available measurements to provide a view of two distinct aerosol pollution events. However, they fail to use the information to extract something new that would be of value for the scientific community. In*

20 *particular, they miss to apply the relationship obtained in Figure 13 to an independent set of measurements to assess its potential value for other users, cities, or pollution events.*

*Overall, the study appears to be driven solely by the availability of data rather then a genuine scientific question. This is particularly obvious in the use of the spaceborne lidar measurements that are not used in a way to provide added value. Also,*

25 *Section 2.1 should not be much larger than the sections on the other instruments. Information from other parameters, such as the Angstrom exponent from the sun photometer measurements, is not exploited sufficiently. For instance, it could be used in Figure 7 and the related discussion to assess if the variation in lidar ratio is related to changes in aerosol size.*

We would like to thank the referee for having taken note of our work. The topic of the air quality is indeed of importance. This paper contributes to the scientific literature bringing an original dataset recorded by lidar during major winter APE which is, as far as we know, only sparsely documented during this season and even less so in the Paris conurbation. In addition, we show that if one wants to assess air quality with lidar in the environmental conditions discuss in the paper, the aerosol extinction coefficient will be far more suitable than integrated value as the aerosol optical thickness. This last conclusion calls into question the methodology using the AOT as a proxy for ground-level PM described in several published works but the present paper does not aim to propose an universal law linking ground-level concentration and optical properties in the PBL. Indeed, one way of effectively and quantitatively assessing and predicting regional air quality using lidar technology is to assimilate lidar data into a chemistry-transport model, which has been studied by Wang et al. (2013, 2014) as explained in the text.

The relation obtained and presented in the Figure 13 is the result of the dataset dispersion in terms of $PM_{2.5}$ and $AEC_{max}$. This kind of dispersion in the data appears when a significant APE is sampled before it starts and until its paroxysm. Such cases are rarely sampled because they are infrequent, difficult to predict and a lidar must be in operation at the same time. Through the WASLIP (Winter Aerosol Survey by Lidar In Paris) field campaign planed one year in advance, we monitored the aerosol pollution in the Paris area by setting up the LAASURS lidar system from 1$^{st}$ November to 31$^{st}$ January in Paris centre. During the experiment, two major APEs occurred during the 2016/2017 winter.

The use of CALIOP in winter for measurements within the PBL is very difficult because of the ground echo, there is a strong uncertainty on the AEC. Moreover, the daytime data are too noisy and the measurement is at 532 nm which would change the spectrally-dependent slope of the linear law in Figure 13.

Section 2.1 is indeed more developed because it presents the central tool of the experiment and the information processing approach which is crucial in lidar signal analysis.

The Angström exponent (AE) has been studied and even used within the lidar inversion process. It is coherent with the fine mode fraction, which is another relevant AERONET product. Furthermore, most of the LR variability occurred during nighttime, preventing the availability of sun-photometer data.

An independent dataset has been added in a 14$^{th}$ Figure following the request on applying the relationship obtained in Figure 13.

Changes have been made in the text:

"**In the Figure 15 a third independent dataset is added to test the relevance of the linear fit shown in Figure 14. As the dataset recorded during the two major APEs, this third dataset is also sampled during the WASLIP experiment. It covers four days from the 5$^{th}$ to 8$^{th}$ December 2016 and the data processing methodology is the same for the three datasets. This third period corresponds with an intermediate pollution situation with $PM_{2.5}$ between 20 and 60 μg m$^{-3}$, included in the range of the two other polluted periods ($PM_{2.5}$ from ~5 to 90 μg m$^{-3}$). Figure 15 shows that this new independent dataset fits pretty well in the 95% prediction interval of the Figure 14 linear regression. Furthermore, the computation of a linear regression on all the points results with a significant Pearson correlation coefficient $R^2 \approx 0.61$. The slope of these regression lines is highly dependent on the chemical composition of the aerosols as shown by Raut**

and Chazette (2007). They correspond to the mean specific cross-section of the sampled aerosols, which is highly variable and a function of the emission sources, but also of the aerosol ageing processes within the atmospheric environment. An approach using remote sensing of aerosols with the choice of a good optical proxies can give an estimate of the surface pollution in terms of $PM_{2.5}$, but should be used with caution. The most promising approach is

5   the direct assimilation of the raw lidar observation into a chemistry-transport model including measurement modelling (Wang et al., 2013, 2014)."

[Figure]

Figure 1: Relationship between $PM_{2.5}$ (X-axis) and $AEC_{max}$ within the PBL (Y-axis). Orange dots are the same dataset presented in Figure 13. Blue crosses are data sampled from the 5 to the 8 December 2016. The orange solid line corresponds to the linear

10  regression computed from the orange points (same as Figure 13) and the light orange area illustrates its 95% confidence interval. The blue solid line is the result of the linear regression calculated on all the points (blue crosses and orange dots). The correlation coefficients can be found in the top left-hand corner.

**Response to the editor comments and notes in the annotated manuscript**

We are very grateful to Editor to have reviewed the manuscript and submitted helpful comments and suggestions to improve

15  both the study and the text. Here we respond point by point.

0) On the question about AERONET Angstrom exponent (in acp-2019-464-comments-to-the-authors p.8):

*how big is the variability of the AERONET AE in Paris during January 20-23? Is this magnitude of AE variability*

20  *similar to that of Dieudonnee et al. (by the way, how big is theirs?)?*

During January 20-23 the AE varies from 0.5 to 1.6 and is superior to 1.4 most of the time (Jan. 20 and 21). This variability 1 ± 0.5 is typically the input incertitude of the Monte-Carlo sensibility analysis that conducted Royer et al (referenced in Dieudonné et al) and result in 4% error on the LR calculation.

    1) P.1 L.1

*Add two before exceptional in the title*. Changes have been made in the title:

"Remote sensing of **two** exceptional winter aerosol pollution events and representativeness of ground-based measurements"

    2) P.1 L.7 to L.17

*Modification of the abstract*. Changes have been made in the text:

"**Two intense winter aerosol pollution events, which took place in winter 2016-2017 in Paris, were monitored using a ground-based N2-Raman lidar, in the framework of WASPLIT (Winter Aerosol Survey by Lidar In Paris), a dedicated field campaign that was carried out in this area from 1st November 2016 to 31st January 2017.** The data analysis uses the synergy between ground-based and spaceborne lidar observations, and data from the air quality monitoring network Airparif. The first severe aerosol pollution event began on 30th November 2016 and ended on 2nd December; concern**ing** a circular area of ~250 km in diameter around Paris. The maximum PM10 was 121±63μg m-3 (**regional** spatial average ± standard deviation) **for the aerial Airparif ground-based PM monitoring stations** and the aerosol extinction coefficient (AEC) ranged from 0.2 to 1 km-1. The second event took place from 20th to 23rd January which covered all of **the** North-western Europe, with maxima of PM10 around 156±33 μg m-3 and AEC between 0.6 and 1 km-1, within the winter atmospheric boundary layer. **Although** these two major aerosol pollution events **did not occur under identical anticyclonic weather conditions, they** share very low planetary boundary layer (PBL) heights, down to 300 m above ground level."

    3) P.1 L.28

*Contributors -> factors*. Changes have been made in the text:

 "of the main anthropogenic **factors**"

    4) P.2 L.17

*With -> by*. Changes have been made in the text:

"often concerned **by** air pollution issues"

    5) P.2 L.26-27

*Dilution capability along the altitude, are low -> Vertical dispersion of pollution, is poor*. Changes have been made in the text:

"They demonstrate that in weather-blocking conditions and with a cold surface, the PBL height **is low, and therefore the vertical dispersion of pollution is poor,** resulting in forecast uncertainties"

    6) P.2 L.27

*to obtain increased vertical and temporal resolution -> which enable a high vertical resolution*. Changes have been made in the text:

"lidar remote sensing techniques **which enable a high vertical** resolution can provide valuable data"

    7) P.2 L.33

*underlie the establishment -> prevail during the two*.Changes have been made in the text:

"that **prevail during the two** significant winter APEs"

    8) P.2 L.34

*to characterise […] to investigate*. Changes have been made in the text:

"**to** characterize observed APEs using in situ and remote sensing data and finally **to** investigate the link"

    9) P.3 L.7

*link cannot be made between "ground based aerosol measurements" and "particles trapped within the winter PBL". As stated in the title, the link of ground based aerosol measurements is probably going to be made with columnar properties of aerosols, within PBL. Please correct*.

Changes have been made in the text:

"**to** investigate the link between ground-based aerosol **concentrations** and **optical properties of** particles trapped within the winter PBL."

    10) P.3 L.10-11-12

*retrieved […] **along** with […] to assign -> to identify*. Changes have been made in the text:

": i) lidar measurements are inverted in order to **retrieve** aerosol optical properties, such as the lidar ratio (LR) and the aerosol extinction coefficient (AEC), ii) the linear particle depolarization ratio (LPDR) is then assessed and used **along** with the LR to identify the aerosol typing, iii) the meteorological situation is examined **to identify** the origin of pollution aerosol over the Paris area"

    11) P.4 L.24 to 29

*Make a comment to support the use of a constant Angstrom Exponent*. Changes have been made in the text (section 2.1.2 and 2.1.3):

"In the inversion process, […] **furthermore a mean value of the Ångström exponent is taken for each APE**. […] The assumption of constant exponent in **altitude** is consistent as all aerosols are concentrated in a well-mixed shallow PBL. **In addition, AERONET data shows a temporal variability during each APE below ±0.5**. […] **The constant Ångström exponent assumption with an input incertitude of ±0.5 will induce uncertainties of 4% on the LR and 1.5% on the AEC as calculated by Royer et al. (2011a)**."

    12) P.8 L.14

*daily -> daily mean*. Changes have been made in the text:

"on daily **mean** PM values"

    13) P.8 L.16

*perform a spatial average on all the background stations -> compute spatial averages over all the background stations*.
Changes have been made in the text.

"we **compute** spatial average**s over** all the background stations"

     14) P.8 L.19

5   *Could not be -> could be not*. Changes have been made in the text:

"could be **not** representative"

     15) P.8 L.20

*Add "during the eleven winters from 2007/2008 to 2017/2018"*. Changes have been made in the text:

"the information threshold **during the eleven winters from 2007/2008 to 2017/2018**."

10     16) P.8 L.26

*unify the two sentences*. Changes have been made in the text:

"Yet, winter 2016/2017 stands out with a large number of threshold **exceedances, in opposite to** the previous and following winters, i.e. 2015/2016 and 2017/2018, **that** present few threshold exceedances."

     17) P.8 L.27

15  *Sentence re-written and references added*. Changes have been made in the text:

**This multi-annual variability is related with the prevailing meteorological conditions, namely the occurrence of strong and persisting anticyclonic situations**. Indeed, despite a general trend in emissions to decline in the Paris region, there are still noteworthy episodes of pollution. When a strong high pressure system sets in over a long period of time, it prevents air mass advection, blocking the weather situation. Thus, the pollution still emitted, even if it is less than in the past, remains

20  blocked by the high pressure system and ends up exceeding the health thresholds **(Menut et al., 1999a; Vautard et al., 2003; Chazette and Royer, 2017)**.

     18) P.9 L.18 (Table 2 legend)

*Add "winter" before days*. Changes have been made in the text:

"The 8 most severely polluted **winter** days"

25     19) P.12 L.1 to 4 (Figure 3 legend)

*Maps should be inverted and the legend has to state the data origin*. Changes have been made in the text and maps has been inverted:

"(a) **Weather situation on 1ˢᵗ December 2016 at 12:00 UTC and (b) on 22ⁿᵈ January 2017 at 12:00 UTC taken from ECMWF ERA-5 reanalysis.**"

30     20) P.13 L.3 (Figure 4 legend)

*Add a mention on the color scale of the windrose*. Changes have been made in the text:

"**The coloured scale for wind speed (Ws) refers to the windrose.**"

21) P.14 L.15

*Except on -> except for*. Changes have been made in the text:

"the sun-photometer-derived AOT **except for** 1st December"

22) P.14 L.17

*is this information* [on the presence of clouds identified on lidar vertical profiles] *taken from analyses performed during the campaign. Are they shown somewhere? If so, there should be referenced, if not they should be accompanied by "personal communication"*

Changes have been made in the text:

"These discrepancies are mainly due to the presence of middle and high-altitude clouds identified on lidar vertical profiles **(white bands in the figure 6b and 8a)**, which may bias the AERONET operational products (Chew et al., 2011)."

23) P.14 L.21

*Reformulation of the sentence and add of the year for each dates*. Changes have been made in the text:

"Note that for 1st December **2016** (22nd January **2017**), when comparing the MODIS-derived AOT at 550 nm of 0.12±0.07 (0.15±0.07) with the AOT$_{phot}$ of 0.16±0.06 (0.11±0.03) at the same wavelength, **their difference of 0.04 is within** the error bars."

24) P.16 L.1

*These temporal variations -> this temporal variation*. Changes have been made in the text:

"**This** temporal **variation**".

25) P.16 L2

*On -> in*. Changes have been added in the text:

"as highlighted **in** Figure 8b"

26) P.16 L.5-6

*Resuspensions and reformulation of the sentence*. Changes have been made in the text:

"Indeed, mechanical processes inducing abrasion (tires, breaks …) linked to human activity and **resuspension** processes**, which are the main source of coarse particles with aerodynamic diameters larger than 2.5 µm**, decrease during the night."

27) P.16 L.11-15

*all this discussion about the possible role of RH for the rise of LR is rather confusing and eventually unnecessary. First, it should be clearly stated that RH rises during the night (because of the temperature drop). For hygroscopic aerosols, this would mean a stronger growth and larger particle sizes. This would act contrary to the reported smaller particle size during night. Yet, the documented weak role of RH, reported by the authors, suggests to better remove the text referring to RH as an explanatory factor for smaller night particle sizes.*

The discussion on the possible role of RH was a request from the referee #3 in his report during the initial manuscript evaluation.

"8) Lidar ratio diurnal cycle (section 4.2): the authors ascribe this pattern to the diurnal pattern of urban emissions; however they could also be linked to the diurnal cycle of RH (driven by T) if the aerosols are hygroscopic. I think it is worth commenting in this."

For the moment, we choose to follow your suggestion and omit this discussion. If it is deemed necessary to discuss the role of RH on the LR, this text seems appropriate to us:

"**The possible role of RH diurnal cycle has been examined. It is worth noting that the increasing RH during nighttime due to temperature drop argues in favour of an increase in particle size and would act contrary to the reported smaller particle observed at night. Furthermore, the RH measured by radiosondes and modelled by ERA5 does not exceed 60%, close to the liquefaction point of aerosol in Paris (Randriamiarisoa et al., 2006). We therefore do not believe that the influence of RH on LR is significant, as also demonstrated during the LISAIR field campaign by Raut and Chazette (2007) for RH < 80%.**"

28) P.16 L.14

*77±6% -> 0.77±0.06*. Changes have been made in the text: "**0.77±0.06**"

29) P.16 L.28

*We assume -> it can be assumed*. Changes have been made in the text:

"Thus, **it can be assumed** that the spaceborne and ground-based lidars measured the same type of aerosol and that their data are comparable."

30) P.17 (Figure 7 legend)

(a) and (b) replaced at the beginning of each sentence.

31) P.17-18 (Figure 7 and 8)

Indication of the orange shaded area with a rectangle in the label.

32) P.19 (Figure 9)

*Indicate, using symbols, on the map figure the exact location of the cities, with emphasis to that of Paris (e.g. using bigger size of symbol).*
Crosses have been added to the map with a bigger one for Paris.

33) P.19 (Table 3)

*PDR -> LPDR*. The correction have been made.

34) P.20 L.14-17

*Divided by two -> halved ; and -> when ; dates instead of Figure number*. Changes have been made in the text:

"Indeed, as seen in Figure 7b, the PBL height is **halved** during 21$^{st}$ January **when** PM$_{10}$ doubles. The standard deviation of both PM$_{2.5}$ and PM$_{10}$ are larger on **30$^{th}$ November and 1$^{st}$ and 2$^{nd}$ December 2016 than on 20$^{th}$, 21$^{st}$ and 22$^{nd}$ January 2017**."

35) P.22 L.5-11

*This is an important conclusion, which however needs a physical explanation/support behind it. Please provide such an explanation.* Changes have been made in the text:

[revised manuscript text omitted]

---

## Author Response (AR3)

*Remote sensing of two exceptional winter aerosol pollution events and representativeness of ground-based measurements*

*by A. Baron, P. Chazette and J. Totems*

**Authors response to the referee #5 and Editor**

5 Dear Editor,

Please find hereafter the response to referee #5. The authors thank you for thoughtful and constructive insights on the manuscript. We appreciate the time you invested in the review. We believe that our revised manuscript addresses all the comments; there is also an enhancement of the results regarding the added independent dataset with which we test the relationship of Figure 13.

10 Kind regards,

Alexandre

In the following, *reviewer comments are in italic blue*. Responses are in normal black font. Changes in the manuscript are highlighted in the paper inserted after this response.

**Authors response to the Editor**

*(i) please address the concerns of Reviewer#5*

Please see our response hereafter.

*(ii) emphasize the applicability and limitations of the derived relationship from the 2 extreme pollution*
20 *cases with respect to the weaker 3rd event. You should do this in the Abstract, where now only the two last sentences refer to generalized findings and not to descriptions and reports of measured quantities, as well as in Conclusions where two sentences in the last paragraph (lines 7-10) do so.*

Additions have been made in the abstract and the conclusion as well as in section 5. Please see the highlighted changes in the manuscript.

*(iii) discuss the slopes of scatterplots in Figure 14, also including the slope of the 3rd dataset in this Figure*

A section 5.3 named "discussion" has been added to address the request for an application of the presented technique to an independent dataset. In this section the slopes are discussed, and the discussion is reliably

5  supported by added data from a separate period in December.

**Authors response to the referee #5**

*The authors have addressed only one of my concerns, which is the application to an independent data set. However, the results are not convincing. [...]*

*Instead, the authors just added the comparison of maximum extinction coefficient versus measured PM*
10  *to Figure 15. In addition, they don't provide a fit of just the third data set (blue crosses) in new Figure 15. The new fit still includes all the points from the previous cases, and thus, is not an independent evaluation. Even worse, this fit is likely to be very different from the one of cases 1 and 2.*

Following the editor's request, we previously addressed this main concern, but also the other concerns in a document called "Author's Response". Although some of the comments were not referring to
15  specific points as it is customary, which could have helped us correct them, and sometimes a matter of judgement, we tried to give the best responses we could.

In response to this new comment, we have provided the editor with a regression on a separate dataset (5 to 8 December), which first was not conclusive. Further work has been done to understand the mismatch of the independent dataset. It was found that the points which were clearly out of the
20  regression were on the upper part of the PBL top. In the following figure (Figure R1a) one can see that the points between 40 and 50 µg m$^{-3}$ and below 0.4 km$^{-1}$ of AEC (top panel, see the colorbar associated to the record date) are sampled during the periods when the PBL was the most shallow (mornings of 6 and 7 December, Figure R1b). This was leading to a regression physically non-significant and based on data intervals too small to be statistically reliable.

[Figure]

Figures R1 a & b

To enlarge the independent dataset which suffers from a poor range of PM and AEC values, we added more available days and re-compute the lidar inversion lower in altitude than initially (300 to 250 m AMSL, see Figures R2b that can be compared with Figure R1b). The results are significantly improved
5  as all the maximum values of AEC are taken in the PBL and are well correlated with the ground-level PM. In Figure R2a, one can see that the regression performed on the extended dataset (3 to 10 December) confirms the results derived from the regression on the APEs dataset regarding the specific extinction cross-section (9.4 $m^2$ $g^{-1}$).

The computation on the joined datasets show results of the same magnitude (9.3 $m^2$ $g^{-1}$). Yet these specific
10  cross-sections are higher than what can be found in the literature and in particular comparatively to the results of Raut and Chazette (2009) (table 1, maximum of 7.1 $m^2$ g-1) and their review in table 2. These higher values could be explained by the use of $PM_{2.5}$ in this study rather than $PM_{10}$, and the same computation with $PM_{10}$ for the dataset of major APEs leads to a specific extinction cross-section of 7.0 $m^2$ $g^{-1}$ acknowledging this assumption.

15  All these statements have been added in a new discussion part of the paper (see section 5.3).

[Figure]

Figures R2 a & b

*To really show that their results are of merit, the authors need to provide a time series (as in Figure 11)*
5   *that includes the lidar-derived PM values or a correlation plot of measured PM versus lidar-derived PM*
*for periods that have not been used to obtain the lidar conversion factors in the first place.*

The aim of the paper is not to convert aerosol optical characteristics to PM but more to show the
coherence between the PBL lidar observation and the ground-based PM2.5 measurements. Deriving PM

from lidar has already successfully been done in the previous literature. The following are examples of such studies in which one of the present authors have participated:

- Raut, J.-C. and Chazette, P.: Assessment of vertically-resolved PM10 from mobile lidar observations, Atmos. Chem. Phys., 9, 8617-8638, https://doi.org/10.5194/acp-9-8617-2009, 2009.

5        - Royer, P., Chazette, P., Sartelet, K., Zhang, Q. J., Beekmann, M., and Raut, J.-C.: Comparison of lidar-derived PM10 with regional modeling and ground-based observations in the frame of MEGAPOLI experiment, Atmos. Chem. Phys., 11, 10705-10726, https://doi.org/10.5194/acp-11-10705-2011, 2011

We have plotted hereafter the calculation of lidar-derived PM estimates in Figures R3 a, b and c for each period, the two major APEs and the control dataset. However, we do not think that these figures belong
10 in the paper because they are redundant with the scatterplots.

[Figure]

[Figure]

Figures R3 a, b & c

*Overall, I still don't think that the manuscript is of a quality that would be sufficient for publication in ACP. While there is scientific significance to the work in principle, the manuscript does not present substantial new concepts, ideas, methods, or data when compared to earlier work on the topic.*

This kind of major polluted event is very scarcely documented in the literature and even less so by lidar remote sensing. The data is difficult to obtain due to the rarity of such events, and the dataset is completely new. The investigation of the coherence between in-situ and remote sensing in such circumstances is new as well and is very relevant in the broader context of studying particulate pollution with remote sensing networks or space-based sensors, which is a growing trend. We thus strongly object to this subjective comment. Note that one of the authors is a senior expert in lidar applied to the environment.

The scientific significance of the paper has been assessed in the first place by 4 referees, even before entering in the discussion phase, which is not customary. All of them stated a "good". Indeed, if the paper was not in the scope of or relevant for publication in ACP, it should not have been accepted for interactive discussion.

*Data from a few weeks of new measurements is not sufficient to fulfil those criteria. Right now it is not clear if the obtained relationship can be generalised to other periods of high PM load or for other highly-polluted sites.*

As we are dealing with unprecedented, continuously recorded lidar data during APEs, there are indeed some difficulties to test this method on other datasets. These are very rare events, as we have shown in the paper, obtaining all available in-situ data in the area for the past 10 years. In section 5 of the paper, we find an optical parameter derived from lidar measurements which could be a proxy of the PM2.5 concentration at the ground-level at a regional scale (AEC lidar in the atmospheric column right above it versus PM averaged over all the Paris Area). The linear fits show that lidar measurements match with ground-based PM measurements when considering the aerosol extinction coefficient within the PBL. This shows that the monitoring of such extreme pollution events by lidar is relevant if one uses the advantage of its vertical resolution, and not integrated values.

Albeit the finite nature of our dataset (15 days of data altogether), we believe that the physical explanation drawn in the 5th paragraph is robust and trustworthy. It is quite clear, as explained in the text of the article, that the nature of aerosols does not change much between extreme pollution events and that the study presented is therefore significant. Parisian emissions mainly related to traffic will not change from one year to the next and the lidar and PMs measurements will show the same type of relationship.

Therefore, we think that the argument stating that our dataset is too short is not relevant, as it is simply the only one in existence. The reviewer may not be used to deal with datasets originating from field campaigns rather than permanent stations, but they are always limited in time and it often happens, for a

specific physical process for example, that only a small interesting subset of the collected data is displayed and studied in a research article (including numerous papers edited by ACP).

*In addition, the presentation quality is still low due to redundant text and an excessive use of figures when put in relation to the amount of text. I would recommend that the authors submit their work to a*
5  *different, maybe more specialised journal."*

This comment is vague: we would benefit from the reviewer's indications about where the redundancies are. Also, the use of figures allows the reader to get a snapshot of the observations and results, a very important element of scientific papers. They are the evidence on which the conclusions are based. This article is no richer than others in terms of figures. Finally, this seems incoherent with the reviewer's own
10  request for two other figures to illustrate the PM-lidar correlation on another dataset and on a time series.

[revised manuscript text omitted]

---

## Author Response (AR4)

**Remote* sensing of two exceptional winter aerosol pollution events and representativeness of ground-based measurements**

**by A. Baron, P. Chazette and J. Totems**

**Authors response to the referee #5 and Editor**

5 Dear Editor,

Please find hereafter our response to your comments. We appreciate the time you invested in the review. Kind regards,

Alexandre

10 2. Please make the requested modifications shown in section 5.3 of the annotated document.

Concerning the manuscript, I made the corrections you asked in section 5.3 and updated the few links to figures that were obsolete.

In the new Figure 14, it seems that the group of points in previous Figure 14 with PM values between
 40 and 50 μg m-3 and AECs below 0.4 km-1, have disappeared in the new Figure 14, where the new/extended dataset seems to now fit very well, and also to provide a very similar slope to that of the two major pollution events. Please, explain how this was achieved. In fact, the third dataset has been extended, from 05-08/10/2016 to 03-10/12/2016, thus the group of points would be expected to be enriched (more days/points) but also to include the previous (05-08/12) group of points.

20 Also, relevant with this change, please correct the caption of figure 14 referring to the third dataset, i.e. blue crosses, and modify the third sentence referring to the time period.

It is true that we have more points than in the previous version but no points have been removed. Here is the explanation of what has changed with the points between 40 and 50  $\mu$ g m-3 and AECs below 0.4 km-1 that were in the previous version of figure 14:

We pointed out in the last response that these points which looked really uncorrelated were indeed points corresponding to times where the PBL was very shallow (see the time-series of the bottom panel of the figure 1).

As a consequence, we think that the corresponding AEC of these points were under-evaluated because taken in the entrainment layer or above the PBL, and thus no more surface-correlated. The enhancement of the lidar inversion permitted to retrieve AEC lower in the PBL (300 m to 250 m AMSL). And it appears that the maxima of the AEC found in the PBL for these profiles were found lower in altitudes and larger in AEC values.

To conclude, these points are still there in the new version and one can relies on the correspondence
with PM2.5 to be sure of that. The only change is on the shift in AEC that has been permitted by an inversion of the lidar signal lower in altitude (one can see the difference in the times-series of figure 1 and figure 2).